# Dynamics of transcriptional programs and chromatin accessibility in mouse spermatogonial cells from early postnatal to adult life

Irina Lazar-Contes[1,2†], Rodrigo G Arzate-Mejia[1,2†], Deepak K Tanwar[1,2†], Leonard C Steg[1,2], Kerem Uzel[1,2], Olivier Ulrich Feudjio[3], Marion Crespo[3], Pierre-Luc Germain[1,2], Isabelle M Mansuy[1,2]*

[1]Laboratory of Neuroepigenetics, Brain Research Institute, Medical Faculty of the University of Zurich and Institute for Neuroscience, Department of Health Science and Technology of the ETH Zurich, Zurich, Switzerland; [2]Center for Neuroscience Zurich, ETH and University Zurich, Zurich, Switzerland; [3]ADLIN Science, Pépinière «Genopole Entreprises», Evry, France

**\*For correspondence:**
mansuy@hifo.uzh.ch

[†]These authors contributed equally to this work

## eLife Assessment

This **useful** study reports datasets on gene expression and chromatin accessibility profiles of spermatogonia at different postnatal ages in mice. Overall, the technical aspects of the sequencing analyses and computational/bioinformatics are **solid**. This study may be of interest to biomedical researchers working on male germline stem cells and male fertility.

**Abstract** In mammals, spermatogonial cells (SPGs) are undifferentiated male germ cells in testis that are quiescent until birth and then self-renew and differentiate to produce spermatogenic cells and functional sperm from early postnatal life throughout adulthood. The transcriptome of SPGs is highly dynamic and timely regulated during postnatal development. We examined if such dynamics involves changes in chromatin organization by profiling the transcriptome and chromatin accessibility of SPGs from early postnatal stages to adulthood in mice using deep RNA-seq, ATAC-seq and computational deconvolution analyses. By integrating transcriptomic and epigenomic features, we show that SPGs undergo massive chromatin remodeling during postnatal development that partially correlates with distinct gene expression profiles and transcription factors (TF) motif enrichment. We identify genomic regions with significantly different chromatin accessibility in adult SPGs that are marked by histone modifications associated with enhancers and promoters. Some of the regions with increased accessibility correspond to transposable element subtypes enriched in multiple TFs motifs and close to differentially expressed genes. Our results underscore the dynamics of chromatin organization in developing germ cells and complement existing datasets on SPGs by providing maps of the regulatory genome at high resolution from the same cell populations at early postnatal, late postnatal and adult stages collected from single individuals.

## Introduction

Spermatogonial cells (SPGs) are the initiators and supporting cellular foundation of spermatogenesis in testis in many species, including mammals. In the mammalian testis, the founding germ cells are

primordial germ cells (PGCs), which give rise sequentially to different populations of SPGs: primary transitional (T1)-prospermatogonia (ProSG), secondary transitional (T2)-ProSG and then spermatogonial stem cells (SSCs) (*McCarrey, 2013*; *Rabbani et al., 2022*; *Tan et al., 2020*). The ProSG population is exhausted by postnatal day (PND) 5 (*Drumond et al., 2011*) and by PND6-8, distinct SPGs subtypes can be distinguished on the basis of specific marker proteins and regenerative capacity (*Cheng et al., 2020*; *Ernst et al., 2019*; *Green et al., 2018*; *Hermann et al., 2018*; *Tan et al., 2020*).

SSCs represent an undifferentiated population of SPGs that retain regenerative capacity and divide to either self-renew or generate progenitors that initiate spermatogenic differentiation, giving rise to differentiating SPGs (diff-SPGs). Diff-SPGs form chains of daughter cells that become primary and secondary spermatocytes around PND10 to 12. Spermatocytes then undergo meiosis and give rise to haploid spermatids that develop into spermatozoa. Spermatozoa are then released into the lumen of seminiferous tubules and continue to mature in the epididymis until becoming capable of fertilization by PND42-48 in mice (*de Rooij, 2017*; *Kubota and Brinster, 2018*; *Oatley and Griswold, 2017*).

Recent work showed that SPGs in early postnatal life have distinct transcriptional signatures (*Green et al., 2018*; *Hammoud et al., 2014*; *Hermann et al., 2018*; *Law et al., 2019*). During the first week of postnatal development, SPGs display unique features necessary for the rapid establishment and expansion of the cell population along the basement membrane. This includes high expression of genes involved in cell cycle regulation, stem cell proliferation, transcription, and RNA processing (*Grive et al., 2019*). In comparison, genes expressed in adult SPGs are involved in the maintenance of a balance between proliferation and differentiation and help constitute a steady cell population that ensures sperm formation across life. Previous transcriptome analyses have revealed that adult SPGs prioritize pathways related to paracrine signaling and niche communication, as well as mitochondrial functions and oxidative phosphorylation (*Grive et al., 2019*; *Hermann et al., 2018*). In addition, changes to chromatin accessibility, histone tail modifications and DNA methylation have also been reported in SPGs during postnatal development (*Maezawa et al., 2018*; *Hammoud et al., 2014*; *Hammoud et al., 2015*). Today however, the relationship between transcriptome dynamics and chromatin landscape in SPGs during the transition from early postnatal to adult stage has not been fully characterized.

We characterized the transcriptome and chromatin accessibility in SPGs during the transition from early postnatal to adult stages in mice. The results reveal extensive changes in transcription and chromatin accessibility in particular, increased accessibility at enhancer regions in adult compared to postnatal SPGs. Regions with changes in chromatin accessibility are enriched in binding motifs of several TFs that are differentially expressed between postnatal and adult stages, suggesting a possible role in changes in chromatin accessibility. Analyses of chromatin accessibility at transposable elements (TEs) identified previously uncharacterized changes at long terminal repeats (LTR) and LINE L1 subtypes between developing and adult SPGs. Together, these findings suggest a functional link between transcriptional dynamics and chromatin accessibility in SPGs during development and underscore the plasticity of genome organization in male germ cells.

## Results
### FACS enriches SPGs collected from postnatal and adult mouse testis

We collected testes from 8- and 15-day-old pups (PND8 and 15) and from adult males and prepared cell suspensions from each animal by enzymatic digestion. The preparations were enriched for SPGs by fluorescence-activated cell sorting (FACS) using specific surface markers (*Kubota et al., 2004*; *Figure 1A–B*). The purity of sorted cells was evaluated by immunocytochemistry using PLZF (ZBTB16), a well-established marker for SPGs (*Costoya et al., 2004*). FACS enriched cell populations from 3–6% PLZF + (before FACS) to 85–95% PLZF+ (*Figure 1B*), suggesting high SPGs enrichment.

To characterize the molecular identity of enriched SPGs populations, we profiled their transcriptome at PND8, PND15 and adulthood by total RNA sequencing (RNA-seq) (n=6 for each group) and examined the expression of known SSCs and somatic markers (Leydig and Sertoli cells) (*Figure 1C*). Classical SPGs markers such as *Kit* (*Schrans-Stassen et al., 1999*), *Id4* (*Helsel et al., 2017*; *Sun et al., 2015*), *Lin28a* (*Chakraborty et al., 2014*; *Wang et al., 2020*), *Zbtb16* (*Costoya et al., 2004*; *Song et al., 2020*), and *Gfra1* (*He et al., 2007*) were robustly expressed in all SPGs samples at each developmental stage (*Figure 1C*), indicating high purity of sorted cells. In contrast, low to negligible

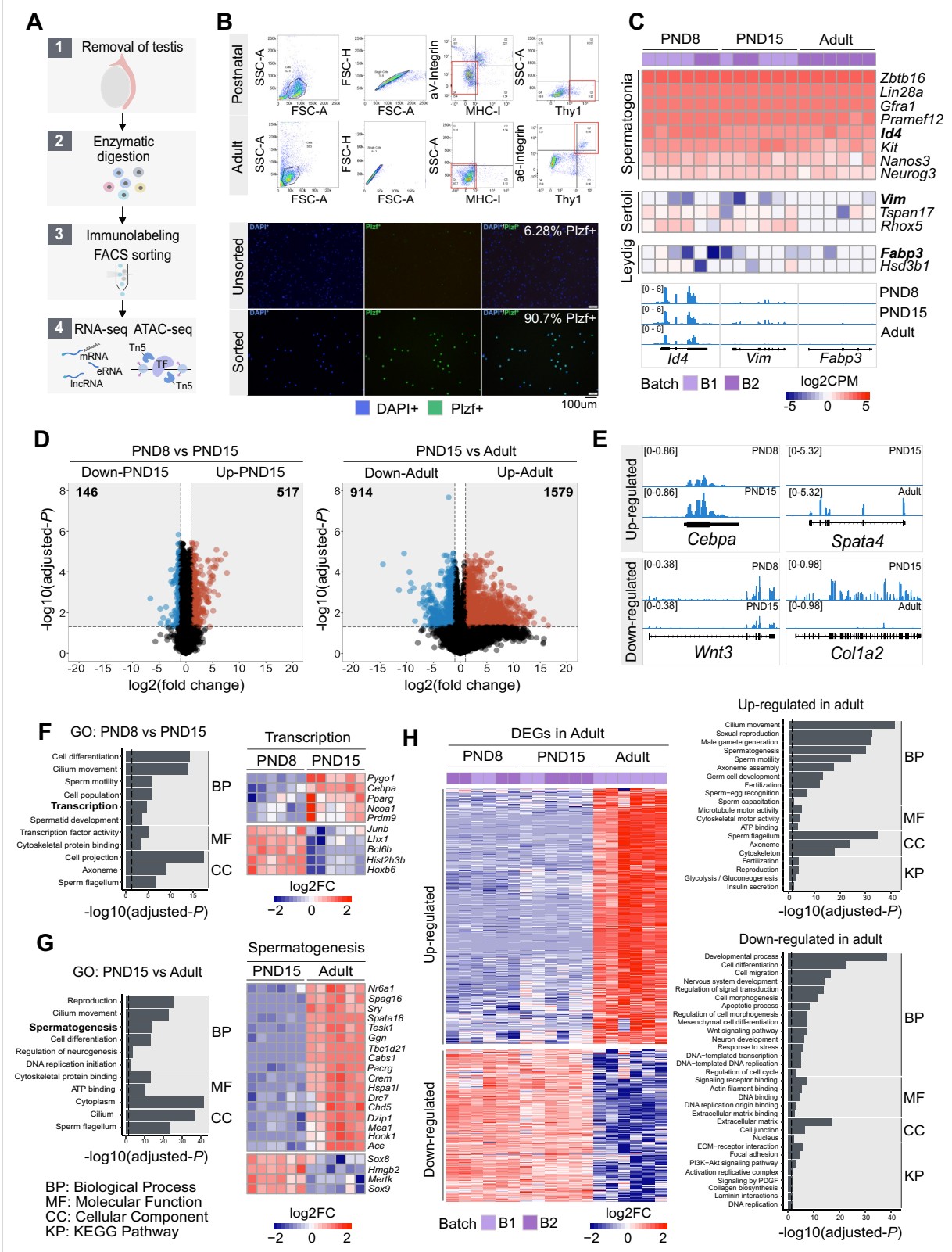

**Figure 1.** Transcriptome dynamics between early postnatal and adult SPGs. (**A**) Schematic representation of the experimental strategy to isolate and analyse SPGs from postnatal or adult male mice. (**B**) Upper panel: Representative dot plots of the sorting strategy for enrichment of postnatal and adult SPGs. Gating based on side scatter/forward scatter (SSC-A/FSC-A) and forward scatter (height/forward scatter) (FSC-H/FSC-A) area was conducted to exclude cell debris and clumps. Lower panel: Representative immunocytochemistry on unsorted and sorted PND15 cells with anti-

*Figure 1 continued on next page*

*Figure 1 continued*

PLZF antibody and DAPI (nuclei) showing enrichment of PLZF + cells after FACS. (**C**) Heatmap of expression profile of selected markers of SPGs and different testicular somatic cells extracted from total RNA-seq data on PND8, PND15 and adult samples (n=6 for each group). Key genes for stem cell potential, stem and progenitor spermatogonia and Leydig and Sertoli cells were chosen to assess the enrichment of SPGs in the sorted cell populations. Each row in the heatmap represents a biological replicate from two experimental batches (B1 and B2). At the bottom Integrative Genomics Viewer (IGV), tracks are shown for aggregated RNA-seq signal for PND8, PND15 and adult over key genes used as markers. Gene expression is represented as Log2CPM (counts per million). (**D**) Volcano plot of differentially expressed genes (DEGs) (adjusted-p ≤0.05 and absolute Log2FC ≥ 1) between PND8 and PND15 (left) and PND15 and adult (right). The numbers in the grey boxes indicate the number of down- and up-regulated DEGs in each comparison. (**E**) Genomic IGV snapshots of exemplary DEGs showing aggregated RNA-seq signal for PND8, PND15 and adult SPGs. (**F**) Left: Bar-plot of GO categories enriched in DEGs between PND8 and PND15 (adjusted-p ≤0.05). Dotted line indicates threshold value for significance of 0.05. Right: Heatmap of DEGs between PND8 and PND15 belonging to the GO category 'Transcription'. Each row represents a biological replicate from two experimental batches. Shown are Log2FC with respect to average at PND8. (**G**) Left: Bar-plot of GO categories enriched in DEGs between PND15 and adult (adjusted-p ≤0.05). Dotted line indicates threshold value for significance of adjusted-p=0.05. Right: Heatmap of DEGs between PND8 and PND15 belonging to the GO category 'Spermatogenesis'. Each row represents a biological replicate from two experimental batches. Shown are Log2FC with respect to average at PND8. (**H**) Left: Heatmap of all DEGs specific to adult SPGs. Each row represents a biological replicate from two experimental batches (B1 and B2). Log2FC with respect to average at PND8 is shown. Right: Bar-plots of GO categories enriched in up-regulated (adjusted-p ≤0.05) (top) or down-regulated (adjusted-p ≤0.05) (bottom) genes in adult SPGs. Dotted line indicates threshold value for significance of adjusted-p=0.05.

The online version of this article includes the following figure supplement(s) for figure 1:

**Figure supplement 1.** Expression of SPGs markers and their dynamics between early postnatal and adult stage.

**Figure supplement 2.** Deconvolution analyses.

**Figure supplement 3.** Specific transcriptional programs between postnatal and adult SPGs.

expression of somatic cells markers such as *Vim* (*Bernardino et al., 2018*) and *Tspan17* (*Gewiss et al., 2021*) for Sertoli cells, and *Fabp3* and *Hsd3b1* for Leydig cells (*Sararols et al., 2021*; *Figure 1C*) was detected. To further validate the samples purity, we manually curated a list of spermatogonial and somatic cell markers derived from recent single-cell RNA-seq datasets (*Cao et al., 2021*; *Sararols et al., 2021*) and determined their expression level in our datasets. Again, we detected robust and consistent expression of all reported spermatogonial markers and low expression of somatic cells markers in all samples at each developmental stage (*Figure 1—figure supplement 1A*).

To evaluate the cellular composition of samples, we applied computational deconvolution to our bulk RNA-seq datasets, employing publicly available single-cell RNA-seq datasets (*Cobos et al., 2023*). Trained on high-quality RNA-seq datasets from pure or single-cell populations, deconvolution algorithms create expression matrices reflecting the cellular diversity in reference datasets. These cell-type-specific expression matrices are subsequently used to determine the cellular composition of bulk RNA-seq samples. First, we assessed the cellular composition of all our RNA-seq libraries, using datasets generated by *Hermann et al., 2018*, which characterized the single-cell transcriptomes of various populations of SPGs and testicular somatic cells in early postnatal (PND6) and adult stages. This enabled us to analyze the composition of isolated SPGs but also to assess potential somatic cell contamination using a unified dataset source. Upon re-analyzing single-cell RNA-seq datasets, we identified distinct cell-type clusters, marked by specific cellular markers as reported in the original and subsequent studies (*Figure 1—figure supplement 2A–E*). Then, CIBERSORTx generated gene-expression signature matrices and estimated the cell-type proportions within our 18 bulk RNA-seq libraries (*Newman et al., 2019*). Evaluation of our postnatal libraries (PND8 and PND15) against a PND6 signature matrix revealed a predominant derivation from SPGs, with average estimated proportions of SSCs being 0.99 and 0.85 for PND8 and PND15 samples, respectively (*Figure 1—figure supplement 2F*). Notably, the analysis of PND15 libraries also suggested the presence of additional SPGs types, including progenitors and differentiating SPGs (*Figure 1—figure supplement 2F*), albeit at lower frequency. Similarly, evaluation of our adult RNA-seq libraries using an adult signature matrix, showed an average SSCs proportion of 0.82, indicating a primary derivation from SSCs (*Figure 1—figure supplement 2G*). Consistent with the findings from PND15 libraries, our deconvolution analysis also suggests the presence of additional SPGs types, including progenitors and differentiating SPGs in adult samples (*Figure 1—figure supplement 2G*). However, unlike in postnatal libraries, the deconvolution analysis of adult libraries indicated the presence of other cell types (labeled 'Other'), not corresponding to the major somatic cell types identified by *Hermann et al., 2018*. The estimated average proportion of these cells was less than 0.05 in two adult libraries and 0.10 in the others. This

variance in cellular composition underlines the effectiveness of the deconvolution method to dissect the complex cellular composition of bulk RNA-seq samples. To further strengthen our observations, we re-analyzed two additional testicular single-cell RNA-seq datasets derived from an early postnatal (PND7; *Tan et al., 2020*) and adult (*Green et al., 2018*) stage (*Figure 1—figure supplement 2C–E*). Evaluation of our libraries confirmed a derivation from germ cells, in particular SSCs (*Figure 1—figure supplement 2G–F*). Therefore, the results of the deconvolution analyses and our immunofluorescence data showing 85–95% PLZF + cells in our cellular preparations confirm the efficiency of our FACS-based enrichment method and underscore that our bulk RNA-seq libraries are mainly composed of SPGs. The deconvolution analyses also suggest a predominant cellular composition of SSCs and to a lesser degree of differentiating SPGs, while also detecting a small proportion of somatic cells (<0.10) in our adult RNA-seq libraries, which is expected given the use of an immunolabeling-based enrichment method.

## Dynamic transcriptomic states characterize postnatal and adult SPGs

We examined the transcriptome dynamics of SPGs from postnatal to adult stages by identifying differentially expressed genes (DEGs) in the RNA-seq datasets. We used a stringent cut-off (adjusted-p ≤0.05, abs $Log_2$ fold change (FC) ≥1) on 17,000 genes expressed during at least one developmental stage (see Methods). A total of 663 DEGs were identified between PND8 and PND15 (146 down-regulated and 517 up-regulated) and 2483 DEGs between PND15 and adult stage (914 down-regulated and 1579 up-regulated) (*Figure 1D–E* and *Supplementary file 1*). Consistent with previous reports (*Hammoud et al., 2015*), we observed a dynamic regulation of germ cell factors, transcription factors (TF) involved in core pluripotency pathways and signaling molecules important for self-renewal (*Figure 1—figure supplement 1B*). For instance, *Pou5f1 (Oct4)*, a TF necessary for pluripotency (*Nichols et al., 1998*), is significantly down-regulated in adult SPGs while the TFs *Klf4* and *Sox2*, also needed for pluripotency (*An et al., 2019*), are expressed similarly in postnatal and adult stages although at different level (i.e. *Klf4* is expressed more than *Sox2*; *Figure 1—figure supplement 1B*).

We conducted Gene Ontology (GO) enrichment analyses to identify the biological processes which DEGs are involved in. GO analyses showed that DEGs between PND8 and PN15 are involved in cell differentiation, cilium movement, sperm motility and transcription and include for instance, TFs such as *Junb*, *Hoxb6*, *Cebpa*, and *Pparg* (*Figure 1F* and *Supplementary file 2*). Further, genes coding for histone proteins such as histone H2B *Hist2h3b* and epigenetic modifiers like the methyltransferase *Pygo1* and histone acetyltransferase *Ncoa1* are also differentially expressed between PND8 and PND15. Interestingly, genes down- or up-regulated between postnatal stages are involved in different processes. While down-regulated genes are implicated in cell differentiation, cell migration and regulation of proliferation for instance, *Cdkn1a* that is down-regulated at PND15 regulates genetic diversity during spermatogenesis (*Kanatsu-Shinohara et al., 2022*), up-regulated genes are rather involved in germ cell development, cell signalling, insulin secretion, and transcription (*Figure 1—figure supplement 1C* and *Supplementary file 2*). Further, DEGs between PND15 and adulthood play roles in reproduction, spermatogenesis, cell differentiation, DNA replication, glycolysis and extracellular matrix (ECM)-receptor interaction pathways (*Figure 1G*, *Figure 1—figure supplement 1D*, and *Supplementary file 2*). For example, the DNA methyltransferase *Dnmt1*, necessary for epi/genome regulation (*Edwards et al., 2017*) and *Col4a2*, a subunit of type IV collagen and component of the basal membrane (*Reissig et al., 2019*), are specifically down-regulated in adult SPGs (*Figure 1—figure supplement 1C*). Interestingly, expression of collagen has been associated to a high proliferative potential and the ability to form germ cell colonies in SPGs (*He et al., 2005*), suggesting that regulation of collagen genes in adult SPGs may decrease germ-stem cell potential. Genes up-regulated in adult SPGs are involved in cilium movement, germ cell development and glycolysis (*Supplementary file 2*).

We then examined the differences and similarities of transcriptional profiles across the three developmental stages. To identify unique or common changes in gene expression during the transition from early postnatal to adult stages, we compared the list of DEGs at PND8 versus PND15 and at PND15 versus adulthood. Remarkably, the vast majority of DEGs show significant stage-specific changes in transcription (*Figure 1—figure supplement 3A*). For instance, 75% (495/663) of DEGs between PND8 and PND15 are not statistically changed in the transition to adult SPGs (*Figure 1—figure supplement 3B*, and *Supplementary file 1-Supplementary file 2*). Similarly, 93% (2325/2493)

of DEGs in adult SPGs are not statistically changed when compared to PND8 versus PND15 SPGs (*Figure 1H*, and *Supplementary file 1-Supplementary file 2*). GO enrichment analyses showed that adult-specific down-regulated genes are involved in cell differentiation, cell migration, regulation of signal transduction and Wnt signalling, regulation of DNA replication and transcription as well as collagen biosynthesis and laminin interactions while adult-specific up-regulated genes are involved in sexual reproduction, male gamete generation, germ cell development, cilium movement and glycolysis (*Figure 1H*, and *Supplementary file 1-Supplementary file 2*).

Interestingly, a small fraction of all DEGs (4.9%) are detected as significantly changed in the same direction and magnitude (adjusted-p ≤0.05, absLog$_2$FC ≥1) across the three developmental stages (*Figure 1—figure supplement 3C*, and *Supplementary file 1-Supplementary file 2*). For instance, 121 genes are consistently up-regulated from PND8 to adult stage, while just 26 are consistently down-regulated from PND8 to adult stage (*Figure 1—figure supplement 3C*). Such DEGs are involved in different biological pathways like chromatin organization (*Supplementary file 2*). In particular, the histone gene clusters displayed significant down-regulation across postnatal development and in adulthood (*Supplementary file 2*).

## Landscape of chromatin accessibility in SPGs during postnatal development

Cellular differentiation is generally accompanied by changes in chromatin accessibility at regulatory elements (*Atlasi and Stunnenberg, 2017*). We examined how chromatin accessibility in SPGs is modified during development using omni-ATAC-seq (*Corces et al., 2017*). We focused on SPGs at PND15 and adulthood, stages that showed the highest changes in gene expression (*Figure 1*). Accessible regions were identified by peak-calling on merged nucleosome-free fragments (NFF) as proxy for genomic regions with potential regulatory activity. We identified 158,977 peaks with clear ATAC-seq signal compared with surrounding genomic regions (*Figure 2—figure supplement 1A*, and *Supplementary file 3*). Most accessible regions were located at distal intergenic regions (38%), introns (28%) and putative promoter regions (23%; *Figure 2—figure supplement 1B–C*) and encompassed sequences enriched for histone PTMs associated with active chromatin such as H3K4 mono-, di-, and tri-methylation (*Figure 2—figure supplement 1D*). Notably, signal enrichment was higher for H3K4 methylation than for other histone PTMs (*Figure 2—figure supplement 1D*). These results are consistent with previous observations that ATAC-seq peaks are identified at regulatory elements such as enhancers and promoters and are preferentially located at genomic regions with nucleosomes carrying H3K4me (*Henikoff et al., 2020*; *Bleckwehl et al., 2021*).

We then examined differences in chromatin accessibility between PND15 and adult SPGs by differential accessibility analysis. 3212 differentially accessible regions (DARs) were identified (adjusted-p ≤0.05, absLog$_2$FC ≥1) with a total of 760 regions with decreased chromatin accessibility (DARs-down) and 2452 regions with increased accessibility (DARs-up) in adult SPGs (*Figure 2A–B* and *Supplementary file 3*). DARs were predominantly localized in distal intergenic regions (54% DARs-down, 37% DARs-up) and introns (33% DARs-down, 36% DARs-up) with a minority located in putative promoter regions (8% DARs-down, 12% DARs-up), consistent with the genomic distribution of all detected ATAC-seq peaks (*Figure 2C*). GO analyses of the closest genes assigned to each DAR showed that these genes are involved in male gonad development, cell adhesion, sex differentiation and regulation of cell communication among others (*Figure 2D–E*).

Epigenomic annotation of DARs using high-quality publicly available ChIP-seq datasets for postnatal SPGs (*Cheng et al., 2020*) revealed that DARs are predominantly located in regions enriched for histone PTMs associated with enhancers and promoters. 51% of regions with increased accessibility are located at genomic loci that carry histone PTMs since early postnatal stages, and about half (48%) overlap with regions significantly enriched in H3K4me1, a histone PTM associated to enhancers. In contrast, 80% of regions with decreased accessibility are located in regions with H3K4me1 and 1/3 (33%) also overlap with the repressive histone PTM, H3K27me3 (*Figure 2F–G*). Interestingly, while 16% of regions with increased accessibility are located at potential active regulatory elements that carry H3K27ac, none of the regions with decreased accessibility overlap with H3K27ac (*Figure 2F–G*). Therefore, our data indicate that the transition from postnatal to adult SPGs is accompanied by discrete, yet robust, changes in chromatin accessibility at potential regulatory elements, suggesting their involvement in the control of gene transcription.

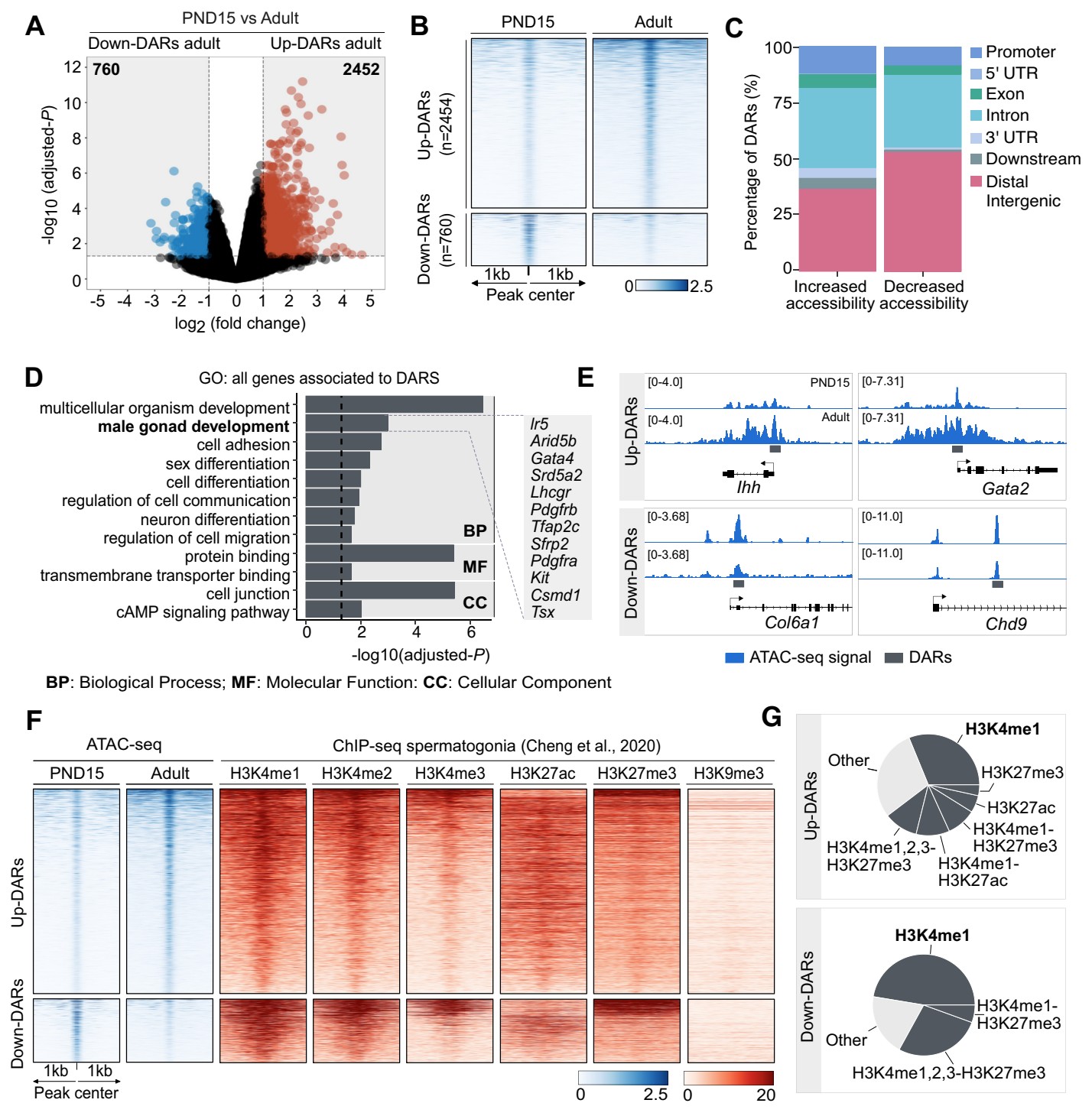

**Figure 2.** Dynamics of chromatin accessibility of SPGs during the transition from postnatal to adult stage. (**A**) Volcano plot of DARs obtained by ATAC-seq (adjusted-p ≤0.05 and absolute Log2FC ≥1) between PND15 and adult SPGs. The numbers in the grey boxes indicate the number of DARs with decreased (Down) or increased (Up) chromatin accessibility in PND15 compared to adult SPGs. (**B**) Heatmaps showing normalized ATAC-seq signal for all identified DARs comparing PND15 and adult stages. Each row represents a 2 kb genomic region extended 1 kb down- and upstream from the centre of each identified DAR (as shown in A). Rows are ordered in a decreasing level of mean accessibility. (**C**) Bar plot illustrating the genomic distribution of DARs between PND15 and adult SPGs. (**D**) Bar plot of GO categories for associated genes assigned by proximity to all DARs. Dotted line indicates threshold value for significance of adjusted-p=0.05. (**E**) IGV tracks for ATAC-seq signal for PND15 and adult SPGs showing DARs (grey box) located at promoters and intergenic regions. (**F**) Heatmaps showing normalized ChIP-seq signal for different histone marks in adult SPGs (public data derived from *Cheng et al., 2020*) at all identified ATAC-seq peaks from PND15 and adult stages. Each row represents a 2 kb genomic region extended 1 kb down-

*Figure 2 continued on next page*

Figure 2 continued

and up-stream from the centre of each identified peak. Shown data correspond to non-regenerative SPGs as stated in *Cheng et al., 2020*. (**G**) Pie charts showing the overlap of all identified DARs with genomic regions significantly enriched for different histone marks in adult SPGs (public data derived from *Cheng et al., 2020*).

The online version of this article includes the following figure supplement(s) for figure 2:

**Figure supplement 1.** Chromatin accessibility landscape of SPGs from PND15 to adult stage.

## DARs are associated with binding sites for distinct families of TFs

We examined the relationship between differences in chromatin accessibility and TF binding by TF motif analysis using DARs as input. We observed that regions with decreased accessibility in adult SPGs are enriched for binding motifs of members of specific TF families such as KLF (KLF2, KLF5, KLF11, KLF12, KLF15, KLF16), SP (SP1-5, SP9), FOXO (FOXO1, FOXO3), ETS (ETV1-6) among others (*Figure 3A* and *Supplementary file 4*). Expression analyses showed that many of these TFs are differentially expressed in postnatal or adult SPGs (adjusted- p<0.05, 10 down-regulated and 11 up-regulated; *Figure 3B–C*). For example, KLF11,12 and 15, TFs that regulate stem cell maintenance and development (*Bialkowska et al., 2017*), are upregulated in adult SPGs compared to PND15 (*Figure 3C*). These TFs and their motif match top enriched motifs detected in DARs in adult SPGs (*Figure 3A*). The SP family of TFs also show differential expression and enrichment of binding motifs in regions with decreased accessibility. SP5, which has its lowest expression in adult SPGs (*Figure 3B–C*), promotes self-renewal of mESCs by directly regulating *Nanog* (*Tang et al., 2017*).

Members of the FOXO family of TFs also have an enrichment of motifs in regions with decreased accessibility. While FOXO3 has increased expression from PND15 onwards, FOXO1 is decreased in adult compared to postnatal SPGs (*Figure 3B–C*). Interestingly, FOXO1 and FOXO3 regulate SSC function and maintenance in mouse (*Goertz et al., 2011*). Finally, ETS-type of TFs also show differential expression during SPGs development and their motif is enriched in regions with decreased accessibility. ELK4 is up-regulated in adult SPGs and can act as a transcriptional repressor via recruitment of the HDAC Sirt7 and deacetylation of H3K18ac (*Figure 3B–C and E*; *Barber et al., 2012*). EKL4 can also act as transcriptional activator of immediate early genes such as *Fos* (*Dalton and Treisman, 1992*). Interestingly, *Fos* itself codes for a TF important for the regulation of proliferation (*He et al., 2008*) and shows a trend towards up-regulation in adult SPGs (*Supplementary file 1*). In contrast, GABPA is downregulated in adult SPGs compared to PND15 and has been involved in the regulation of proliferation in mESCs (*Ueda et al., 2017*).

Regions with decreased chromatin accessibility also show an enrichment for the binding motif of the TF DMRT1. *Dmrt1* is progressively repressed during SPGs postnatal development and is the lowest in adult SPGs (*Figure 3B–D*). DMRT1 can act either as a repressor or activator and controls testis development and male germ cell proliferation (*Zhang et al., 2016*). DMRT1 can inhibit meiosis and promote mitosis in SPGs by repressing Stra8 (*Matson et al., 2010*). Consistently, we observed transcriptional repression of *Stra8* in adult SPGs (*Supplementary file 1*).

Next, we identified motifs overrepresented in regions with increased chromatin accessibility in adult SPGs. The identified motifs correspond to binding motifs of members of specific TF families such as POU (POU2F2, POU1F1, POU5F1), RFX (RXF1-7), DMRT (DMRT1, DMRT3, DMRTA1-2, DMRTC2), SOX (SOX2, SOX3, SOX5, SOX13), NFY (NFYA, NFYB, NFYC), and AP-1 (JUN, JUNB, JUND, FOS, ATF3, BATF) (*Figure 3F* and *Supplementary file 4*). A subset of them is also differentially expressed (adjusted-p <0.05, 13 down-regulated and 5 up-regulated). The most overrepresented motif is similar to the binding motif of the POU family of TFs, which are critical regulators of stem cells (*Nichols et al., 1998*; *Wu et al., 2010*). *Pou1f1* and *Pou5f1* are transcriptionally repressed in adult SPGs while *Pou2f2* is maximally expressed in PND15 SPGs and down-regulated in adult cells (*Figure 3G–H*). We also identified motifs for members of the RFX family of TFs, which are master regulators of ciliogenesis (*Choksi et al., 2014*) implicated in regulation of neural stem cells (*Kawase et al., 2014*). RFX2 is robustly expressed in adult SPGs (*Figure 3G–H and J*) and has been reported to induce the expression of ciliary genes in association with the TF FOXJ, which has a trend towards up-regulation in adult SPGs (*Quigley and Kintner, 2017*). Interestingly, ciliary genes are among the top genes specifically up-regulated in adult SPGs (*Figure 1H*), suggesting a regulatory relationship between RFX2 and ciliary genes expression in adult SPG cells.

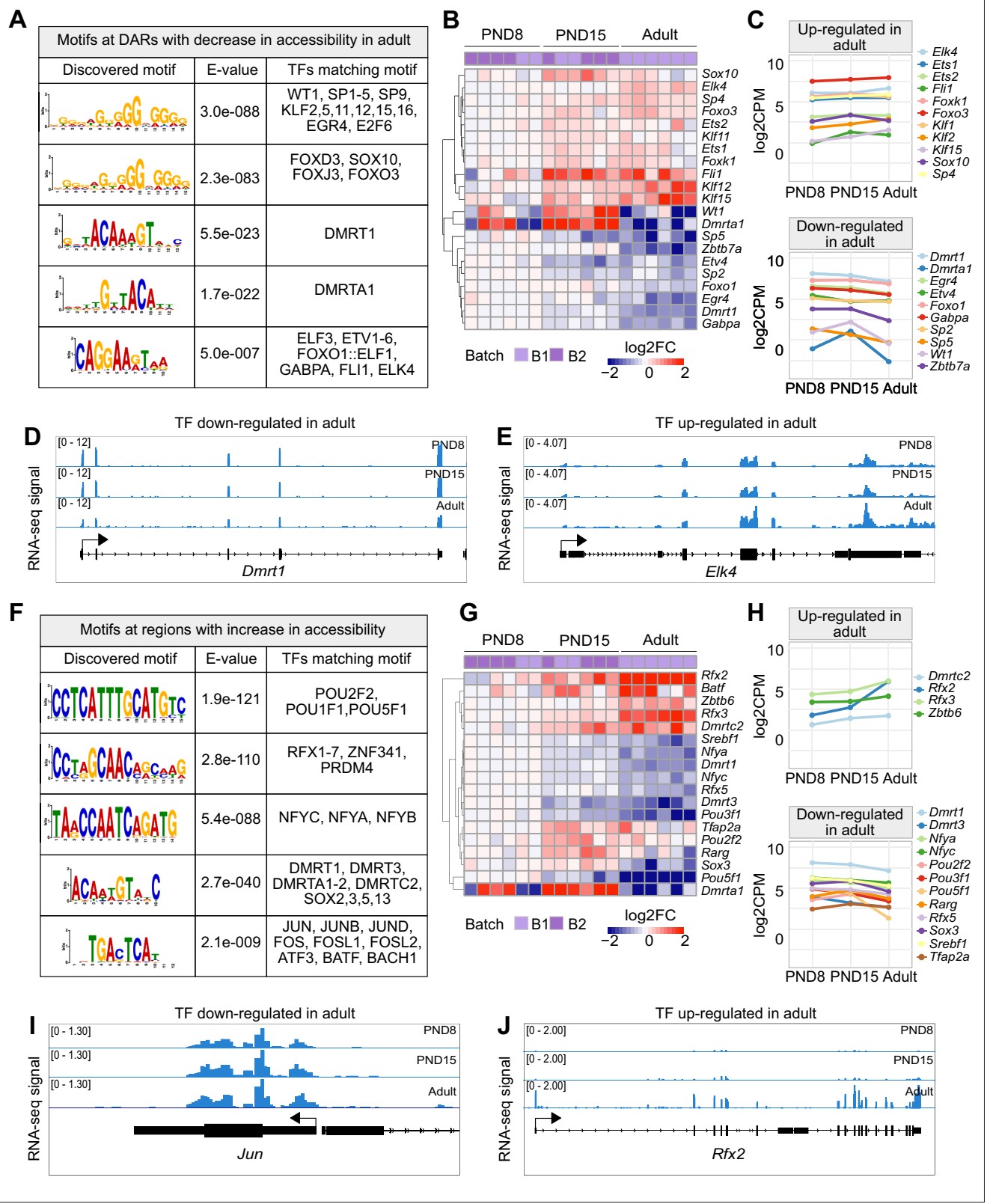

**Figure 3.** TF binding motifs at DARs and their transcriptional dynamics during the transition from postnatal to adult SPGs. (**A**) Table of top five enriched motifs in genomic regions with decrease in chromatin accessibility in adult SPGs and TF matching motifs. A full list of TF motifs is provided in *Supplementary file 4*. (**B**) Heatmap of expression profiles of TFs with motifs in genomic regions with decreased chromatin accessibility shown in (A). Each row represents a biological replicate from two experimental batches. Unsupervised hierarchical clustering was applied to each row and a

*Figure 3 continued on next page*

*Figure 3 continued*

dendrogram indicating similarity of expression profiles among genes is shown. Log2FC is shown with respect to average of PND8 SPGs. (**C**) Line-plots of average expression of each gene displayed in the heatmap in (B) showing the dynamics of gene expression across PND8, PND15 and adult SPGs. (**D, E**) Genomic snapshot from IGV showing aggregated RNA-seq signal from PND8, PND15 and adult SPGs for the TFs (D) *Dmrt1* and (E) *Elk4*. (**F**) Table of top five enriched motifs in genomic regions with increase in chromatin accessibility in adult SPGs and TF matching motifs. A full list of TF motifs is provided in ***Supplementary file 4***. (**G**) Heatmap of expression profile of TFs with motifs in genomic regions with increase in chromatin accessibility shown in (F). Log2FC is shown with respect to the average of PND8 SPGs. (**H**) Line-plots of average expression of each gene displayed in the heatmap in (G) showing the dynamics of gene expression across SPGs development. (**I, J**) Genomic snapshot from IGV showing the aggregated RNA-seq signal from PND8, PND15 and adult SPGs for the TFs (I) *Jun* and (J) *Rfx2*.

Other binding motifs enriched at regions with increased accessibility correspond to members of the NF-Y complex, NF-YA, NF-YB and NF-YC. In mESCs, NF-Y TFs facilitate a permissive chromatin conformation and play an important role in the expression of core ESC pluripotency genes (***Oldfield et al., 2014***). NF-YA/B motif enrichment has also been found in regions of open chromatin in human SPGs (***Guo et al., 2017***). Interestingly, the expression of NF-YA and NF-YC is progressively down-regulated starting at PND15 (***Figure 3G–H***). We also detected an enrichment for the binding motifs of members of the AP-1 family of TFs, which are involved in many processes, from regulation of cell proliferation to differentiation and acute responses to environmental clues (***Bejjani et al., 2019***; ***Vier-buchen et al., 2017***). Interestingly, except for *Fosl2*, other TFs do not show any significant change in transcription in postnatal or adult SPGs and are either constitutively expressed or repressed. For instance, *Fos* and JUN are robustly expressed in postnatal and adult SPGs (***Figure 3I***) consistent with their role in promoting the proliferative potential of SSCs (***He et al., 2008***; ***Wang et al., 2018***). Together, these data reveal that a shift in TFs repertoire accompanies chromatin reorganization occurring in SPGs during postnatal to adult development.

## A subset of DARs is associated with differential gene expression

Thus far, our results show that in SPGs, the transition from postnatal to adult stage is accompanied by changes in gene expression and chromatin accessibility. In some instances, differences in gene expression can be correlated with differences in chromatin accessibility of regulatory elements. Since DARs in postnatal and adult SPGs overlap with putative enhancer and promoter elements, we examined if these changes in chromatin accessibility correlate with changes in gene expression. First, we tested if promoter regions of DEGs (+/-2.5 kb from TSS) have differential chromatin accessibility by identifying all possible ATAC-seq peaks located in promoters and calculating an average accessibility signal. No significant difference was detected for down-regulated genes in adult SPGs, despite a trend for decreased accessibility around the TSS (***Figure 4A***). However, down-regulated genes had a significant increase in chromatin accessibility in their body (***Figure 4B***). Up-regulated genes had a significant increase in chromatin accessibility both at the promoter region and gene body (***Figure 4A–B***). Then, we conducted motif enrichment analyses and identified families of TFs with overrepresented binding motifs at the promoter of DEGs. Binding motifs for members of KLF and SP family of TFs were over-represented in the promoter of both down- and up-regulated DEGs, while motifs for members of TF families SOX and NFY were detected only in the promoter of down-regulated genes. Promoters of up-regulated genes were specifically enriched in motifs for RFX and AP-1 (***Figure 4C***). These observations suggest that changes in chromatin accessibility, in particular at the promoter of up-regulated DEGs may be associated with differential binding of RFX and AP-1 TFs.

Next, we examined the overlap between DARs and the promoter of DEGs. Unexpectedly, only 3.2% of all promoters of DEGs overlap with at least one DAR (***Figure 4D***). The majority of overlapping DARs (71%) is associated with promoters of up-regulated genes in adult SPGs (***Figure 4D***). For example, *Gpx4*, a peroxidase involved in the control of stemness (***Hu et al., 2021***), is transcriptionally active in adult SPGs and has an open chromatin at its promoter region (***Figure 4E***). Since the majority of DARs are located at distal intergenic regions and have histone PTMs associated with active regulatory elements, we extended our search of their target genes by assigning a large region (+/-100 kb) around each DEG. We observed that a third of DEGs were associated with at least one DAR (***Figure 4F***), with the majority of DARs (69%) overlapping regulatory elements of up-regulated genes. For instance, *Braf* is transcriptionally up-regulated in adult SPGs, which correlates with increased chromatin accessibility at distal regulatory elements (***Figure 4G***). Our results overall show that only a fraction of DEGs is

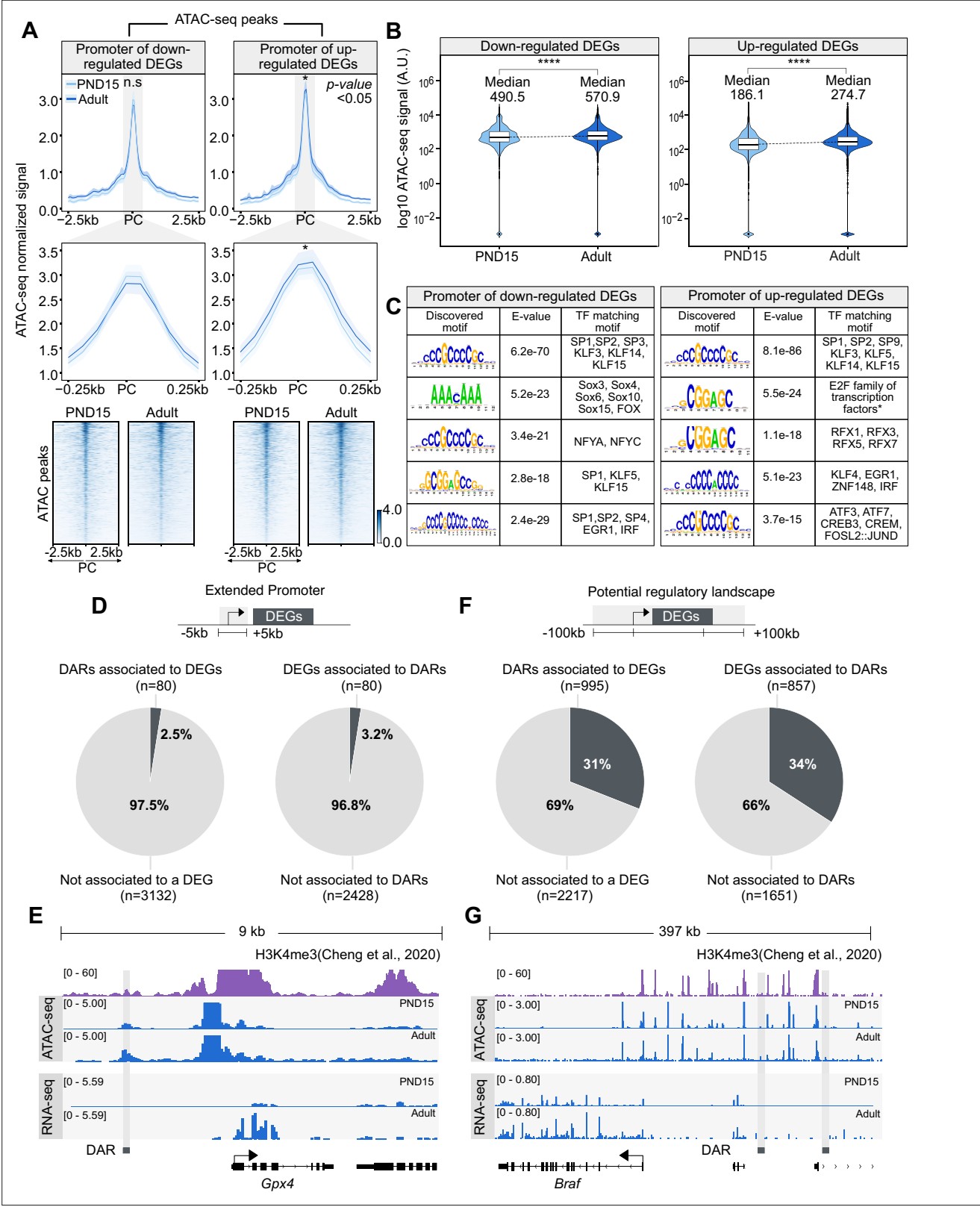

**Figure 4.** Chromatin accessibility dynamics at DEGs during the transition from postnatal to adult SPGs. (**A**) Top: profile plots of ATAC-seq signal from PND15 and adult SPGs over all ATAC-seq peaks located within a genomic region of 5 kb surrounding the TSS of DEGs. Mean (line) and bootstrap confidence interval (area, shadow lines) computed over all non-overlapping 50 bp genomic regions. n.s, non-significant differences at TSS region. *, significant difference in signal between PND15 and adult at p<0.05. PC, peak center from ATAC-seq data. Profile plots for all ATAC-seq peaks

*Figure 4 continued on next page*

*Figure 4 continued*

surrounding the TSS of down-regulated (top left) or up-regulated (top right) DEGs. Middle: profile plots of ATAC-seq signal from PND15 and adult SPGs over all ATAC-seq peaks located within a genomic region of 500 bp surrounding the TSS of DEGs. Bottom: heatmaps showing normalized ATAC-seq signal from PND15 and adult for all ATAC-seq peaks surrounding the TSS of all down-regulated (left) or up-regulated (right) DEGs. Each row represents a 2 kb genomic region extended 1 kb down- and up-stream from the center of each identified peak. Rows are ordered in a decreasing level of mean accessibility taking PND15 as reference. (**B**) Violin plots of average ATAC-seq signal for PND15 and adult SPGs over gene body of down-regulated (left) or up-regulated (right) DEGs. ****p<0.0001. (**C**) TF motif enrichment analysis for all ATAC-seq peaks surrounding the TSS of all down-regulated (left) or up-regulated (right) DEGs. (**D**) Left: pie chart of percentage of DARs overlapping the extended promoter of DEGs (+/-5 kb from TSS). Right: pie chart of percentage of DEGs associated with a DAR. (**E**) IGV tracks for an example of DAR associated to the extended promoter (+/-5 kb from TSS) of a DEG. Tracks for RNA-seq and ATAC-seq signal (blue) are shown for PND15 and adult SPGs generated in this study and signal track for H3K4me3 (purple) from SPGs derived from public ChIP-seq data (*Cheng et al., 2020*). (**F**) Left: pie chart of percentage of DARs overlapping potential regulatory sequences of DEGs (+/-100 kb from start and end of gene). Right: pie chart of the percentage of DEGs associated with a DAR. (**G**) IGV tracks for an example of DAR associated with potential regulatory sequences (+/-100 kb from start and end of gene) of a DEG. Tracks for RNA-seq and ATAC-seq signal (blue) are shown for PND15 and adult SPGs generated in this study and signal track for H3K4me3 (purple) from SPGs derived from public ChIP-seq data (*Cheng et al., 2020*).

associated with DARs, suggesting that changes in gene expression are not always mirrored at the level of chromatin accessibility, consistent with recent observations (*Kiani et al., 2022*).

## Chromatin accessibility at TEs undergoes remodeling during the transition from early postnatal to adulthood in SPGs

TEs are tightly regulated in the germline by coordinated epigenetic mechanisms involving DNA methylation, chromatin silencing and PIWI proteins-piRNA pathway (*Deniz et al., 2019*). SPGs were recently shown to have a unique landscape of chromatin accessibility at long-terminal repeats (LTRs) retrotransposons, compared with other stages of germ cells in testis (*Sakashita et al., 2020*). We examined chromatin accessibility at TEs in PND15 and adult SPGs by quantifying ATAC-seq reads overlapping TEs defined by UCSC RepeatMasker then analyzing differential accessibility at subtypes level (see Methods section). These analyses showed that SPGs transitioning from PND15 to adult stages have significantly different chromatin accessibility at 135 TEs subtypes (adjusted-p ≤0.05, absLog$_2$FC ≥1) (*Figure 5A–B* and *Supplementary file 5*). Most TEs subtypes have decreased chromatin accessibility (93/135, 68,9%) (*Figure 5A*) and 42 have increased accessibility (*Figure 5B*) in adult SPGs. Differentially accessible TEs loci are predominantly located in intergenic (68%) and intronic (25%) regions and 6% re close to a gene (+/-1 kb from TSS; *Figure 5C*).

LTR retrotransposons were the most abundant TEs with altered chromatin accessibility, specifically ERVK and ERV1 subtypes (*Figure 5A–B*). Exemplary ERVK subtypes with less accessible chromatin included RLTR17, RLTR9A3, RLTR12B, and RMER17B (*Supplementary file 5*). Enrichment of RLTR17 and RLTR9 repeats was previously reported in mESCs, specifically at TFs important for pluripotency maintenance such as *Pou5f1* and *Nanog* (*Fort et al., 2014*). Interestingly, we identified the promoter region of the long non-coding RNA (lncRNA) *Lncenc1*, an important regulator of pluripotency in mESCs (*Fort et al., 2014*; *Sun et al., 2018*), to harbor several LTR loci with decreased accessibility in adult SPGs. One of these loci, RLTR17, overlaps with the TSS of *Lncenc1*, and its decreased accessibility correlates with markedly lower *Lncenc1* expression in adult SPGs (*Figure 5D*). *Lncenc1* (also known as *Platr18*) is part of the pluripotency-associated transcript (*Platr*) family of lncRNAs identified as potential regulators of pluripotency-associated genes *Pou5f1*, *Nanog* and *Zfp42* in mESCs (*Bergmann et al., 2015*). We identified several other *Platr* genes such as *Platr27* and *Platr14*, whose TSS overlaps with LTRs with reduced accessibility, RLTR17 and RLTR16B_MM, respectively (*Figure 5D* and *Supplementary file 5*). These two pluripotency-associated transcripts tended to be downregulated in adult SPGs but were unchanged at PND8 and PND15 (*Figure 5D* and *Supplementary file 1*). The other LTR subtypes with decreased accessibility in adult SPGs belong to ERV1, ERVL, and MaLR families (*Figure 5A*). A few non-LTR TEs also had decreased chromatin accessibility, particularly 7 DNA element subtypes, 2 satellite subtypes and 1 LINE subtype, respectively (*Supplementary file 5*).

TE subtypes with increased accessibility mostly included members of ERVK, ERVL and ERV1 families (24/42, 57.1%) (*Supplementary file 5*). Interestingly, many LINE L1 subtypes had increased chromatin accessibility in adult SPGs (*Supplementary file 5*). When parsing the data for more accessible loci within L1 subtypes, we found several L1 loci situated less than +/-5 kb from the TSS of olfactory (*Olfr*) genes, particularly *Olfr* gene clusters on chromosome 2, 7, and 11 (*Supplementary file 5*).

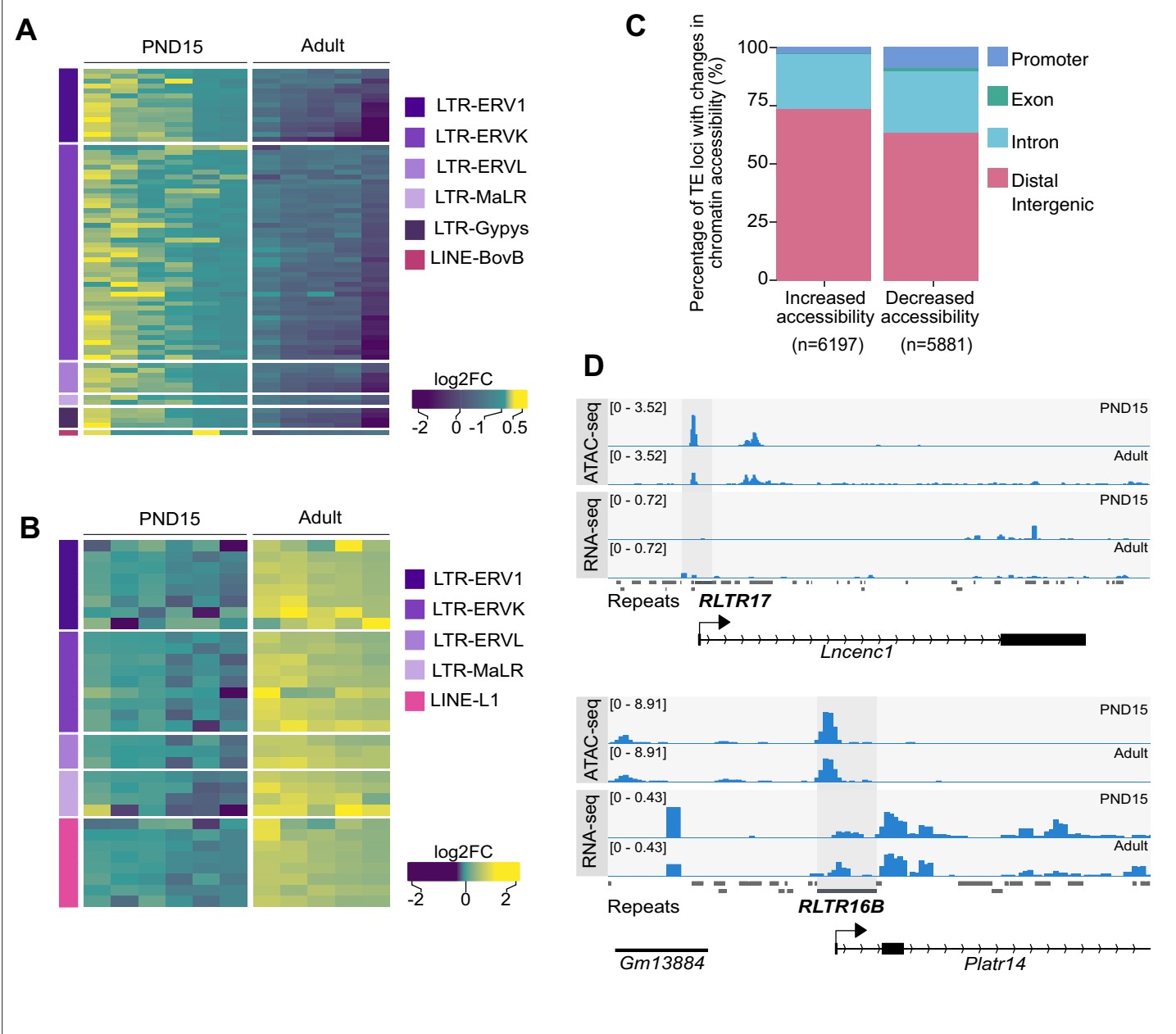

**Figure 5.** Differential chromatin accessibility at TEs in PND15 and adult SPGs. (**A**) Heatmap of LTR and LINE subtypes with decreased accessibility between PND15 and adult SPGs extracted from Omni-ATAC data (adjusted-p ≤0.05 and Log2FC ≥0.5). (**B**) Heatmap of LTR and LINE subtypes with increased accessibility between PND15 and adult SPGs extracted from Omni-ATAC data (adjusted-p ≤0.05 and Log2FC ≥0.5). Expression changes of these subtypes between adult and PND15 SPGs RNA-seq from literature are represented as Log2FC. (**C**) Bar plot illustrating the genomic distribution of DARs between PND15 and adult SPGs. (**D**) Genomic snapshot from IVG showing aggregated ATAC-seq and RNA-seq signal from PND15 and adult SPGs for *Lncenc1* and *Platr14*. The RepeatMasker track for the mouse genome is shown below signal tracks ('Repeats'), and TEs with changes in chromatin accessibility are highlighted.

TEs are known to provide tissue-specific substrates for TF binding (*Sundaram and Wysocka, 2020*). We examined the regulatory potential of differentially accessible LTR subtypes by determining the enrichment of TF motifs in these regions. We grouped LTR subtypes by family (EVK, ERV1, ERVL and ERVL-MaLR families) and observed that among families with less accessible chromatin, ERVKs had the highest number of enriched TF motifs in adult SPGs (*Supplementary file 5*). Top hits included TFs with known regulatory functions in cell proliferation and differentiation such as FOXL1 and FOXQ1, stem cell maintenance factors ELF1, EBF1 and THAP11 and TFs important for spermatogenesis PBX3,

ZNF143 and NFYA/B (*Supplementary file 5*). ERVLs displayed motif enrichment for few TFs, among which ETV2, recently reported to be the SSC factor ZBTB7A and the testis-specific CTCF paralog CTCFL (*Supplementary file 5*; *Green et al., 2018*).

More accessible LINE L1 subtypes were highly enriched in TF motifs, particularly in multiple members of ETS, E2F and FOX families (*Supplementary file 5*). The most significant motifs belonged to SSCs maintenance and stem cell potential regulators FOXO1 and ZEB1, as well as TFs recently associated with active enhancers of the stem-cell-enriched population of SPGs such as ZBTB17 and KLF5 (*Supplementary file 5*; *Cheng et al., 2020*). More accessible ERV1s were also enriched in several TF binding sites, including spermatogenesis-related TFs (PBX3, PRDM1, NFYA/B), hypoxia inducible HIF1A and cytokine regulators STAT5A/B, suggesting different metabolic demand between postnatal and adult stage (*Supplementary file 5*).

## Discussion

This study examines the transcriptomic landscape and chromatin accessibility of mouse SPGs from early postnatal development to adulthood. Using FACS-enriched SPGs populations at PND8, PND15 and adulthood, we show that the transcriptome of SPGs is dynamically regulated during development and that many factors including pluripotency-related TFs and signaling molecules are differentially expressed. This correlates with changes in chromatin accessibility although not all DEGs have altered accessibility. TEs such as LTR retrotransposons have modified chromatin accessibility, which in some cases affects pluripotency-associated lncRNAs. TF motif enrichment in differentially accessible TEs suggests various regulatory roles for SPGs. Our findings complement existing datasets on SPGs by providing parallel transcriptomic and chromatin accessibility maps at high resolution from the same cell populations at early postnatal, late postnatal and adult stages collected from single individuals (for adults).

### Different transcriptional states characterize the postnatal development of SPGs

SPGs are known to undergo genome-wide transcriptional changes during development (*Grive et al., 2019*; *Hammoud et al., 2015*; *Hermann et al., 2018*). The present data complement previous findings by showing that postnatal maturation of SPGs is accompanied by an overall decrease in the expression of pluripotency markers and differential expression of signaling molecules and regulators of cell cycle and spermatogenesis. The high quality and depth of our RNA-seq datasets allowed us to uncover specific transitional states of SPGs postnatal maturation that were not known before. For instance, we observed that while the transition from early to mid-postnatal stage (PND8 to PND15) is accompanied by changes in gene expression, the transition from mid-postnatal to adult stage is more extensive, both in number of regulated genes and magnitude of changes. We also observed a clear trend towards up-regulation of gene expression as postnatal maturation proceeds, that may reflect gradual transcriptional increase for some genes and possibly sharp changes at certain developmental stages for others.

Consistently, many transcriptional regulators and chromatin modifiers are differentially regulated during SPGs development. Different families of TFs including factors with a specific role in SPGs like RFX and DMRT, and factors with more widespread functions across cell types such as members of the AP-1 family are modulated, suggesting the contribution of multiple classes of TFs to SPGs development. Notably, several transcriptional repressors and epigenetic silencers are differentially down-regulated which may explain the marked trend towards transcriptional up-regulation in our datasets (*Figure 1D*). Interestingly, we also observed a global down-regulation of histone genes starting at PND15 (*Supplementary file 1*), which may modify the composition and structure of chromatin and be associated with chromatin openness (*Prado et al., 2017*). Overall, our transcriptional data highlight the yet unappreciated dynamic regulation of TFs and epigenetic/chromatin modifiers in SPGs during postnatal development.

## Chromatin accessibility in SPGs is modified at enhancers during postnatal development

Our results suggest that the transition of SPGs from early postnatal to adult stage is accompanied by changes in chromatin accessibility. In particular, accessibility is increased in regions enriched in H3K4me1, a histone PTM typically associated with primed enhancers, but also to a lesser extent, in regions enriched in H3K27ac, a mark associated with active regulatory elements (*Gasperini et al., 2020*; *Shlyueva et al., 2014*). This suggests that the transition of SPGs from postnatal to adult stage involves changes in the regulatory genome, in particular at primed enhancers. Enhancer priming has recently been proposed as an important mechanism for SPGs differentiation (*McCarrey, 2023*). Differences in chromatin accessibility may be partially explained by differential binding of TFs to regulatory elements (*Klemm et al., 2019*). This may involve at least three non-mutually exclusive events: (1) a change in the abundance of TFs, (2) a change in the epigenetic status of TF binding sites due to different amount and/or activity of epigenetic modifiers, and (3) a change in nucleosome positioning around TF binding sites due to different amount and/or activity of chromatin remodelers (*Arzate-Mejia and Mansuy, 2023*). For instance, CpG methylation and nucleosome positioning can influence TF binding at regulatory elements and affect chromatin accessibility (*Barisic et al., 2019*; *Kaluscha et al., 2022*). The fact that *Dnmt1* is downregulated in adult SPGs concomitantly with differential expression of TFs and chromatin modifiers may explain changes in chromatin accessibility at potential enhancers.

## Modest correspondence between transcription and chromatin accessibility in SPGs

Despite robust changes in gene expression in adult SPGs, only a subset of DARs could be assigned to DEGs. This suggests that transcriptional regulation and chromatin accessibility at regulatory elements are not always linked and can be dissociated, as previously reported (*Chereji et al., 2019*; *Kiani et al., 2022*). This implies that chromatin accessibility may be modified without any effects on transcription and conversely, transcription can be activated or repressed without requiring any change in chromatin accessibility. In turn, it is possible that a repressor can be exchanged with an activator at a regulatory element without detectable consequences. It is also possible that changes in chromatin accessibility have no immediate effects on transcription but reflect an intermediate state that primes a gene or genomic locus for later activation or repression such as observed with developmentally- or experience-dependent primed enhancers (*Marco et al., 2020*; *Rada-Iglesias et al., 2011*). Consistently, the majority of DARs with increased accessibility overlap with regions enriched in H3K4me1, a histone modification characteristic of primed enhancers thought to control future transcriptional responses in different cell types (*Rada-Iglesias et al., 2011*). These regions may be important for responses to external cues in adult SPGs. The apparent lack of correspondence between transcription and accessibility may also be in part technical and due to the stringency of filters we used to assign genomic loci to DARs. We cannot exclude the possibility that modest changes in accessibility (<log2FC1) correspond to changes in TF occupancy. Chromatin accessibility has been suggested to not necessarily be the primary determinant of chromatin-mediated gene regulation (*Chereji et al., 2019*). Such possibility could be addressed by examining genome-wide maps of TFs, cofactors and RNA-Pol II occupancy that would provide more accurate information about the regulatory genome and its relationship with observed gene expression programs.

## Chromatin accessibility at TEs during SPGs maturation

Our results reveal differences in chromatin accessibility and TFs motif landscape at different TEs subtypes between PND15 and adult SPGs. ERVK and ERV1 subtypes were the most abundant categories of TEs that become less accessible in adult SPGs, whilst LINE L1 subtypes gained accessibility. Although the majority of these TEs reside in intergenic and intronic regions, we could detect specific loci belonging to differentially accessible ERVK and LINE L1 subtypes localized nearby TSS of distinct gene families. Notably, ERVKs are known to play an important role in the regulation of mRNA and lncRNAs transcription during spermatogenesis (*Davis et al., 2017*; *Sakashita et al., 2020*). The landscape of chromatin accessibility at LTRs loci in SPGs is known to be unique compared to other mitotic germ cells and meiotic gametes in testis (*Sakashita et al., 2020*).

RLTR17, one of the LTR subtypes with decreased chromatin accessibility in adult SPGs overlaps with the TSS of several downregulated lncRNAs from the *Platr* family. *Platr* genes, including *Lncenc1* and *Platr14* identified in our study, are LTR-associated lncRNAs important for gene expression programs in ESCs (*Bergmann et al., 2015*). Interestingly, RLTR17 has also previously been linked to pluripotency maintenance. In mESCs, it is highly expressed and enriched in open chromatin regions and has been shown to provide binding sites for the pluripotency factors Oct4 and Nanog (*Fort et al., 2014*; *Xiong et al., 2022*). Therefore, we suggest that RLTR17 chromatin organization may play a role in regulating pluripotency programs between early postnatal and adult SPGs.

In contrast to decreased accessibility at LTRs loci, LINE L1 subtype had increased accessibility in adult SPGs. Some of these L1 loci were located close to olfactory receptor genes with upregulated mRNA expression. Recent findings in mouse and human ESCs suggest a non-random genomic localization of L1 elements, specifically at genes encoding proteins with specialized functions (*Lu et al., 2020*). Among them, the *Olfr* gene family was the most enriched in L1 elements (*Lu et al., 2020*). Although their role in SPGs is currently not established, Olfr proteins have been implicated in swimming behaviour of sperm cells (*Fukuda and Touhara, 2006*). Given their dynamic regulation in SPGs, we speculate that *Olfr* genes play additional roles in spermatogenesis, other than in sperm physiology. Together with the high number of enriched TF motifs identified at differentially accessible ERVKs and LINE L1 elements, these data highlight a regulatory role for chromatin organization of TEs in SPGs during the transition from development to adult stage (*Sundaram et al., 2014*; *Sundaram and Wysocka, 2020*).

## Limitations

One limitation of our study is the partial purity of SPGs populations obtained by FACS, due to the use of a purification method that does not fully remove other testicular cells from the samples. This therefore does not exclude the possible influence of contaminating cells on the data and their interpretation. Such influence may explain differences between our datasets and datasets in the literature. Nevertheless, by comparing chromatin landscape in developing and adult SPGs, the study reveals an age-dependent dynamic reorganization of chromatin accessibility in these cells. When integrated with our transcriptomic datasets and published histone PTMs profiles, the data provide novel insight into transcriptome-chromatin dynamics of mouse SPGs from early postnatal life to adulthood with great depth and high resolution. This represents an important resource for future studies on mouse germ cells.

## Methods

### Mouse husbandry

Male C57Bl/6 J mice were purchased from Janvier Laboratories (France) and bred in-house to C57Bl/6 J primiparous females to generate males for the experiments. All animals were kept on a reversed 12 hr light/12 hr dark cycle in a temperature- and humidity-controlled facility with food (M/R Haltung Extrudat, Provimi Kliba SA, Switzerland) and water provided *ad libitum*. Cages were changed once weekly. Animals from two independent cohorts generated separately were used for the experiments.

### Germ cells isolation

Germ cells were isolated at postnatal day (PND) 8 or 15 for RNA-seq and ATAC-seq, and at postnatal week 20 (adult) for ATAC-seq. Testicular single-cell suspensions were prepared as previously described with slight modifications (*Kubota et al., 2004*). For PND8 and PND15 cells, testes from two animals were pooled for each sample while for adults, testes from individual males were used. Pup testes were collected in sterile HBSS on ice. Tunica albuginea was gently removed from each testis, making sure to keep seminiferous tubules as intact as possible. Tubules were enzymatically digested in 0.25% trypsin-EDTA (Thermo Fisher Scientific, Reinach, Switzerland) and 7 mg/ml DNase I (Sigma-Aldrich, Buchs, Switzerland) solution for 5 min at 37 °C. The suspension was vigorously pipetted up and down 10 times and incubated again for 3 min at 37 °C. The digestion was stopped by adding 10% fetal bovine serum (Thermo Fisher Scientific) and cells were passed through a 20µm-pore-size cell strainer (Miltenyi Biotec, Adliswil, Switzerland) and pelleted by centrifugation at 600*g* for 7 min at 4 °C. Cells

were resuspended in PBS-S (PBS with 1% PBS, 10 mM HEPES, 1 mM pyruvate, 1 mg/mL glucose, 50 units/mL penicillin and 50 µg/mL streptomycin) and used for sorting. Adult testes were collected and the tunica was removed. Seminiferous tubules were digested in 1 mg/mL collagenase type IV (Sigma-Aldrich) and then in 0.25% Trypsin-EDTA (Gibco), both times for 8–12 min and in the presence of DNase I solution (Sigma-Aldrich). FBS (Cytiva HyClone) was added to a concentration of 10% and the cell suspension was filtered through a 40 µm-pore-size cell strainer (Corning) and washed with DPBS-S 1% FBS, 10 mM HEPES (Gibco), 1 mM sodium pyruvate (Gibco), 1 mg/mL Glucose (Gibco) and 1 X Penicillin-Streptomycin (Gibco) in DPBS (Gibco). 5 mL of the single cell suspension was then overlayed on 2 mL 30% Percoll (Sigma) and centrifuged with disabled break at 600g for 8 min at 4 °C. The supernatant was removed and cell suspension was washed twice in DPBS-S and used for sorting.

## SPGs enrichment by FACS

For pup testis, dissociated cells were stained with BV421-conjugated anti-β2M, biotin-conjugated anti-THY1 (53–2.1) and PE-conjugated anti-αv-integrin (RMV-7) antibodies. THY1 was detected by staining with Alexa Fluor 488-Sav. For adult testes, cells were stained with anti-α6-integrin (CD49f; GoH3), BV421-conjugated anti-β2 microglobulin (β2M; S19.8) and R-phycoerythrin (PE)-conjugated anti-THY1 (CD90.2; 30H-12) antibodies. α6-Integrin was detected by Alexa Fluor 488-SAv after staining with biotin-conjugated rat anti-mouse IgG1/2 a (G28-5) antibody. Prior to FACS, 1 µg/mL propidium iodide (Sigma) was added to cell suspensions to discriminate dead cells. All antibody incubations were performed in PBS-S for at least 30 min at 4 °C followed by washing in PBS-S. Antibodies were obtained from BD Biosciences (San Jose, United States) unless otherwise stated. Cell sorting was performed at 4 °C on a FACS Aria III 5 L using an 85 µm nozzle at the Cytometry Facility of University of Zurich. For RNA-seq at PND8 and PND15, cells were collected in 1.5 ml Eppendorf tubes in 500 µL PBS-S, immediately pelleted by centrifugation and snap frozen in liquid N2. Cell pellets were stored at –80 °C until RNA extraction. For RNA-seq of adults, 1000 SPGs (MHC I negative, alpha-6-integrin and Thy1 positive) per male were sorted into PBS. For Omni-ATAC at PND15, 25,000 cells were collected in a separate tube, pelleted by centrifugation and immediately processed using a library preparation protocol (*Corces et al., 2017*). For Omni-ATAC in adults, 5000 cells from each animal were collected in a separate tube and processed using the same protocol.

## Immunocytochemistry

The protocol used for assessing SPGs enrichment after sorting was kindly provided by Jon Oatley at Washington State University, Pullman, USA. Briefly, 30,000–50,000 cells were adhered to poly-L-Lysine coated coverslips (Corning Life Sciences, Berlin, Germany) in 24-well plates for 1 hr. Cells were fixed in freshly prepared 4% PFA for 10 min at room temperature then washed in PBS with 0.1% Triton X-100 (PBS-T). Non-specific antibody binding was blocked by incubation with 10% normal goat serum for 1 hr at room temperature. Cells were incubated overnight at 4 °C with mouse anti-PLZF (0.2 µg/ml, Active Motif, Waterloo, Belgium) primary antibody. Alexa488 goat anti-mouse IgG (1 µg/mL, Thermo Fisher Scientific) was used for secondary labelling at 4 °C for 1 hr. Coverslips were washed times, mounted onto glass slides with VectaShield mounting medium containing DAPI (Vector Laboratories, Zurich, Switzerland) and examined by fluorescence microscopy. SPGs enrichment was determined by counting PLZF + cells in 10 random fields of view from each coverslip and dividing by the total number of cells present in the same fields of view (DAPI-stained nuclei). The number of PLZF + and PLZF-cells from the 10 different fields of view was averaged.

## RNA extraction and RNA-seq library preparation

For RNA-seq at PND8 and PND15, total RNA was extracted from sorted cells using AllPrep RNA/DNA Micro kit (QIAGEN, Hilden, Germany). RNA quality was assessed using a Bioanalyzer 2100 (Agilent Technologies, Basel, Switzerland). Samples were quantified using Qubit RNA HS Assay (Thermo Fisher Scientific). RNA sequencing libraries were prepared using SMARTer Stranded Total RNA-Seq Kit v3 (Takara Bio Inc, USA) following the recommended protocol with minor adjustments. For PND8 and PND15 SPGs libraries, RNA was fragmented for 4 min at 94 °C. After reverse transcription, samples were barcoded using SMARTer Unique Dual Index Kit (Takara Bio Inc). Five PCR cycles were run and PCR products were purified using AMPure XP reagent (Beckman Coulter Life Sciences; bead:sample ratio 0.8 X). Ribosomal cDNA was depleted following the protocol and final library amplification was

performed with 12 PCR cycles. Libraries were purified twice using AMPure XP reagent (bead:sample ratio 1 X) and eluted in Tris buffer. Adult SPGs libraries were prepared using the option to start directly from cells (1000 cells as input). Cells were incubated for 6 min at 85 °C and processed as indicated for PND8 and PND15 SPGs. Samples were pooled at equal molarity, as determined in a 100–800 bp window on the Bioanalyzer.

## Omni-ATAC library preparation and sequencing

Chromatin accessibility was profiled from PND15 and adult SPGs. Libraries were prepared according to Omni-ATAC protocol, starting from 25,000 PND15 and 5000 adult sorted SPGs (*Corces et al., 2017*). Briefly, sorted cells were lysed in cold lysis buffer (10 mM Tris-HCl pH 7.4, 10 mM NaCl, 3 mM MgCl2, 0.1% NP40, 0.1% Tween-20, and 0.01% digitonin) and nuclei were pelleted and transposed using Nextera Tn5 (Illumina, Zurich, Switzerland) for 30 min at 37 °C in a thermomixer with shaking at 1000 rpm. Transposed fragments were purified using MinElute Reaction Cleanup Kit (QIAGEN). Following purification, libraries were generated by PCR amplification using NEBNext High-Fidelity 2 X PCR Master Mix (New England Biolabs, Bioconcept AG, Allschwil, Switzerland) and purified using Agencourt AMPure XP magnetic beads (Beckman Coulter) to remove primer dimers (78 bp) and fragments of 1000–10,000 bp. Library quality was assessed on an Agilent High Sensitivity DNA chip using the Bioanalyzer 2100 (Agilent Technologies). Six samples were sequenced from PND15 and five from adult SPGs.

## RNA-seq analysis

### Quality control and alignment

100 bp single end sequencing was performed at the Functional Genomics Center Zurich (FGCZ) using a Novaseq SP flowcell on the Novaseq 6000 platform. Quality assessment of FASTQ files was done using FastQC (*Andrews et al., 2012*; version 0.11.8). TrimGalore (*Krueger, 2015*) (version 0.6.2) was used to trim adapters and low-quality ends from reads with Phred score less than 30 (-q 30) and to discard trimmed reads shorter than 30 bp (--length 30). Trimmed reads were pseudo-aligned using Salmon (*Prado et al., 2017*; version 0.9.1) with automatic detection of the library type (-l A), correcting for sequence-specific bias (--seqBias) and correcting for fragment GC bias correction (--gcBias) on a transcript index prepared for the Mouse genome (GRCm38) from GENCODE (version M18; *Harrow et al., 2012*), with additional piRNA precursors and TEs (concatenated by family) from Repeat Masker as in *Gapp et al., 2020*.

### Downstream analysis

Data analyses and plotting were conducted with R (R Core Team, 2020; version 3.6.2) using packages from The Comprehensive R Archive Network (CRAN; https://cran.r-project.org) and Bioconductor (*Huber et al., 2015*). Pre-filtering of genes was done using the filterByExpr function from edgeR (*Robinson et al., 2010*; version 3.28.1) with a design matrix requiring at least 15 counts (min.counts= 15). Normalization factors were obtained using TMM normalization (*Robinson and Oshlack, 2010*) from edgeR package and differential gene expression analyses were done using limma-voom (*Law et al., 2014*) pipeline from limma (*Ritchie et al., 2015*; version 3.42.2). GO enrichment analyses were performed using g:Profiler (https://biit.cs.ut.ee/gprofiler/gost) querying against *Mus musculus* database.

## Deconvolution analyses

*Hermann et al., 2018*: Raw data were obtained from PND6 and adult sorted ID4-EGFP+SPGs. Using the cluster IDs provided in the supplemental datasets, CIBERSORTx analysis (*Newman et al., 2019*) was performed to estimate the proportion of labeled cell types in each RNA-seq library using the default settings (*G.min*=300, *G.max*=500, q=0.1). For deconvolution analyses, we used the aggregated P6 and adult datasets as references.

*Tan et al., 2020*: Raw data from E18, P2, and P7 were obtained. We applied the reported filtering and normalization parameters, followed by UMAP analysis (*Becht et al., 2019*) using the top 10 principal components (PCs). For clustering, the Seurat package was employed (*Butler et al., 2018*, *dims = 1:10* in *FindNeighbors, resolution = 0.025* in *FindClusters*) and resultant cell counts in each cluster corresponded to those reported in *Tan et al., 2020*. Clusters identified as germ, Sertoli, stroma,

Leydig and peritubular myoid (PTM) cells were validated using specific marker genes as per *Tan et al., 2020*; *Green et al., 2018*; *Hermann et al., 2018*. Cells not classified into these clusters were categorized as 'Other Cells'. Next, CIBERSORTx was utilized to estimate the proportion of these labeled cell types in each RNA-seq library using the default settings ($G.min$=300, $G.max$=500, $q$=0.1). For the deconvolution analyses, two replicate libraries from PND7 were used as references, and their averaged results were reported. Additionally, we performed a re-analysis focusing on re-clustered germ cells. Within this subset, SSCs and differentiating SPGs were identified, using marker genes referenced in *Tan et al., 2020*; *Green et al., 2018 Hermann et al., 2018*.

*Green et al., 2018*: Raw data from adult were obtained. We applied the reported filtering and normalization parameters, followed by UMAP analysis (*Becht et al., 2019*) using the top 10 PCs. For clustering, the Seurat package (*Butler et al., 2018*, *dims = 1:20* in *FindNeighbors, resolution = 0.011* in *FindClusters*). Clusters identified as germ, Sertoli, stroma, Leydig, and peritubular myoid (PTM) cells were validated using specific marker genes as per *Tan et al., 2020*; *Green et al., 2018*; *Hermann et al., 2018*. Cells not classified into these clusters were categorized as 'Other Cells'. Next, CIBERSORTx was utilized to estimate the proportion of these labeled cell types in each RNA-seq library using the default settings ($G.min$=300, $G.max$=500, $q$=0.1). For deconvolution analyses, we used eight seminiferous tubule (ST datasets), three spermatogonia enriched (SPGs datasets) and eight Sertoli enriched (SER datasets) as our references separately, and the averages of each group were reported.

## ATAC-seq analysis

### Quality control, alignment, and peak calling

Paired-end (PE) sequencing was performed on PND15 and adult SPGs on Illumina HiSeq2500 platform at FGCZ. FASTQ files were assessed for quality using FastQC (*Andrews et al., 2012*; version 0.11.8). QC was performed using TrimGalore (*Krueger, 2015*; version 0.6.2) in PE mode (--paired), trimming adapters, low-quality ends (-q 30) and discarding reads shorter than 30 bp after trimming (--length 30). Alignment on GRCm38 genome was performed using Bowtie2 (*Langmead and Salzberg, 2012*; version 2.3.5) with the following parameters: allowing fragments up to 2 kb to align (-X 2000), entire read alignment (--end-to-end), suppressing unpaired alignments for paired reads (--no-mixed), suppressing discordant alignments for paired reads (--no-discordant) and minimum acceptable alignment score with respect to read length (--score-min L,−0.4,−0.4). Using alignmentSieve (version 3.3.1) from deepTools (*Ramírez et al., 2016*; version 3.4.3), aligned data (BAM files) were adjusted for read start sites to represent the center of the transposon cutting event (--ATACshift), and filtered for reads with a high mapping quality (--minMappingQuality 30). Reads mapping to the mitochondrial chromosome and ENCODE blacklisted regions were filtered out. To call nucleosome-free regions, all aligned files were merged within groups (PND15 and adult), sorted and indexed using SAMtools (*Li et al., 2009*; version 0.1.19) and nucleosome-free fragments (NFFs) were obtained by selecting alignments with a template length between 40 and 140 bp in length. Peak calling on NFFs was performed using MACS2 (*Zhang et al., 2008*; version 2.2.7.1) with mouse genome size (-g 2744254612) and PE BAM file format (-f BAMPE).

### Differential accessibility analysis

These analyses were conducted in R (version 3.6.2) using packages from CRAN (https://cran.r-project.org) and Bioconductor (*Huber et al., 2015*). Peaks were annotated based on overlap with GENCODE (version M18; *Harrow et al., 2012*) transcript and/or distance to the nearest TSS (https://github.com/mansuylab/SC_postnatal_adult/blob/deepak/bin/annoPeaks.R). The number of extended reads overlapping in peak regions was calculated using csaw package (*Lun and Smyth, 2016*; version 1.20.0). Peak regions that did not have at least 15 reads in at least 40% of the samples were filtered out. Normalization factors were obtained on filtered peak regions using TMM normalization method (*Robinson and Oshlack, 2010*) and differential analysis on peaks (PND15 versus adult) was performed using Genewise Negative Binomial Generalized Linear Models with Quasi-likelihood (glmQLFit) Tests from edgeR package (*Robinson et al., 2010*) (version 3.28.1). Peak regions with an absLog2FC ≥1 and adjusted-p ≤0.05 were categorized as DARs.

## Downstream analysis

Heatmaps of normalized ATAC-seq signal were created using deepTools with default parameters. Genomic annotation of ATAC-seq peak and DARs and identification of closest gene DARs were performed using ChIPpeakAnno (*Zhu et al., 2010*). GO enrichment analysis of genes associated to DARs was performed using g:Profiler (*Raudvere et al., 2019*) querying against the *Mus musculus* database. For the epigenomic annotation of DARs, publicly available signal files and peak files for all histone marks were used (*Cheng et al., 2020*). Heatmaps of ChIP-seq signal for histone PTMs around ATAC-seq peaks and DARs were generated with deepTools. Overlap between ChIP-seq peaks and DARs was generated using Intervene (*Khan and Mathelier, 2017*). TF motif analysis was performed using MEME-ChIP (*Ma et al., 2014*) and motif identification was done querying identified motifs against JASPAR database (*Castro-Mondragon et al., 2022*). Quantification and identification of differences in chromatin accessibility between promoter regions of DEGs was performed with deep-Stats (*Richard, 2019*). To quantify differences in chromatin accessibility between DEGs, bamscale cov (*Pongor et al., 2020*) was employed using ATAC-seq BAM files as input and genomic coordinates of up-regulated and down-regulated genes. Kruskal-Wallis test was used to test for significant differences in chromatin accessibility between each category of DEGs at postnatal and adult stage.

## Differential accessibility analysis at TEs

TE gene transfer format (GTF) file was obtained from http://labshare.cshl.edu/shares/mhammelllab/www-data/TEtranscripts/TE_GTF/mm10_rmsk_TE.gtf.gz on 03.02.2020. This file provides hierarchical information about TEs: Class (level 1, e.g. LTR), family (level 2, e.g. LTR ->L1), subtype (level 3, e.g. LTR->L1->L1_Rod), and locus (level 4, e.g. LTR->L1 ->L1_Rod ->L1_Rod_dup1). TE loci were annotated based on overlap with GENCODE (version M18) as described above for ATAC-seq peaks. Filtered BAM files (without reads mapping to blacklisted or mitochondrial regions) were used to analyze TEs. Mapped reads were assigned to TEs using featureCounts from R package Rsubread (*Liao et al., 2019*; version 2.0.1) and were summarized to Subtypes (level 3), allowing for multi-overlap with fractional counts, while ignoring duplicates. The number of extended reads overlapping at TE loci were obtained using csaw package (*Lun and Smyth, 2016*; version 1.20.0). Subtypes without at least 15 reads, and loci without at least 5 reads in at least 40% of samples were filtered out. Normalization and differential accessibility analysis were performed as described above. Subtypes which had an absolute Log2FC ≥0.5 and adjusted-p ≤0.05 were categorized as differentially accessible subtypes and loci with an absLog2FC ≥1 and adjusted-p ≤0.05 were categorized as differentially accessible loci. For further downstream data analysis, only differentially accessible loci or differentially accessible subtypes were considered. TF motif enrichment analysis was performed using the marge package (*Amezquita, 2018*; version 0.0.4.9999), which is a wrapper around the Homer tool (*Heinz et al., 2010*; version 4.11.1).

## Acknowledgements

We thank Francesca Manuella and Martin Roszkowski for taking care of animals breeding, Yvonne Zipfel for animal care, Silvia Schelbert and Alberto Corcoba for taking care of animal licenses and lab organization and Niharika Gaur for technical help. We thank Catherine Aquino and Emilio Yángüez from Functional Genomics Center Zurich for support and advice with libraries preparation and sequencing. We are very grateful to Jon Oatley, Melissa Oatley, Tessa Lord and Nathan Law for conceptual advice, hands-on training, and for providing detailed protocols for testis dissection and preparation, and immunocytochemistry of SPG cells. We thank Zuguang Gu for support with heatmaps and Ellen Jaspers for help with VOR preparation. We thank Service and Support for Science IT ( www.s3it.uzh.ch) for computational infrastructure. The Mansuy lab is funded by the University Zürich, the ETH Zürich, the Swiss National Science Foundation grant number 31003 A_175742/1, the National Centre of Competence in Research (NCCR) RNA&Disease funded by the Swiss National Science Foundation (grant number 182880/Phase 2 and 205601/Phase 3), ETH grants (ETH-10 15–2 and ETH-17 13–2), European Union Horizon 2020 Research Innovation Program Grant number 848158, European Union projects FAMILY and HappyMums funded by the Swiss State Secretariat for Education, Research and Innovation (SERI), FreeNovation grant from Novartis Forschungsstiftung, and the Escher Family Fund. RGA-M received an ETH Postdoctoral Fellow grant number 20–1 FEL-28. DKT received a Swiss

Government Excellence Scholarship. The funding agencies were not involved in the study design, data collection, and interpretation, nor in the decision to submit the work for publication to eLife.

## Additional information

### Competing interests

Olivier Ulrich Feudjio: Olivier Ulrich Feudjio is affiliated with ADLIN Science, Pépinière and Genopole Entreprises. The author has no other competing interests to declare. Marion Crespo: Marion Crespo is affiliated with ADLIN Science, Pépinière and Genopole Entreprises. The author has no other competing interests to declare. Isabelle M Mansuy: Guest Editor *eLife*. The other authors declare that no competing interests exist.

### Funding

| Funder | Grant reference number | Author |
|---|---|---|
| Swiss National Science Foundation | 31003A_175742/1 | Isabelle M Mansuy |
| Swiss National Science Foundation | 182880/Phase 2 and 205601/Phase 3 | Isabelle M Mansuy |
| European Union Horizon 2020 Research Innovation Program | 848158 | Isabelle M Mansuy |
| European Union projects FAMILY and HappyMums funded by SERI | | Isabelle M Mansuy |
| FreeNovation grant from Novartis Forschungsstiftung | | Isabelle M Mansuy |
| Escher Family Fund | | Isabelle M Mansuy |
| ETH Postdoctoral Fellowship | 20-1 FEL-28 | Rodrigo G Arzate-Mejia |
| ETH | ETH-10 15–2 | Isabelle M Mansuy |
| ETH | ETH-17 13–2 | Isabelle M Mansuy |

The funders had no role in study design, data collection and interpretation, or the decision to submit the work for publication.

### Author contributions

Irina Lazar-Contes, Conceptualization, Resources, Formal analysis, Validation, Investigation, Visualization, Methodology, Writing – original draft; Rodrigo G Arzate-Mejia, Deepak K Tanwar, Data curation, Software, Formal analysis, Investigation, Visualization, Methodology, Writing – original draft, Writing – review and editing; Leonard C Steg, Investigation, Methodology; Kerem Uzel, Data curation, Software, Formal analysis, Methodology; Olivier Ulrich Feudjio, Data curation, Formal analysis; Marion Crespo, Pierre-Luc Germain, Data curation, Software, Formal analysis, Supervision, Methodology; Isabelle M Mansuy, Conceptualization, Resources, Supervision, Funding acquisition, Investigation, Visualization, Methodology, Writing – original draft, Project administration, Writing – review and editing

### Author ORCIDs

Pierre-Luc Germain ⓘ https://orcid.org/0000-0003-3418-4218
Isabelle M Mansuy ⓘ https://orcid.org/0000-0001-7785-5371

### Ethics

Experiments were run during the active cycle of the animals in conformity with guidelines of the Cantonal Veterinary Office of Zurich and the Swiss Animal Welfare Act (Tierschutzgesetz) and under license number ZH057/15 and ZH083/2018.

Reviewer #1 (Public Review): https://doi.org/10.7554/eLife.91528.3.sa1
Reviewer #2 (Public Review): https://doi.org/10.7554/eLife.91528.3.sa2
Reviewer #3 (Public Review): https://doi.org/10.7554/eLife.91528.3.sa3
Author response https://doi.org/10.7554/eLife.91528.3.sa4

## Additional files

### Supplementary files
MDAR checklist

Supplementary file 1. Differentially expressed genes (DEGs).

Supplementary file 2. Gene ontology analysis of DEGs.

Supplementary file 3. Differentially accessible regions (DARs).

Supplementary file 4. TF motif analysis of DARs.

Supplementary file 5. DARs of repeat sequences.

### Data availability
Sequencing data is available via the EBI BioStudies database within the ArrayExpress collection (RNA-seq: E-MTAB-12721; ATAC-seq: E-MTAB-12722). The code employed for data analysis is available from https://github.com/mansuylab/SC_postnatal_adult (copy archived at *Tanwar, 2025*) and https://github.com/mansuylab/Lazar-Contes2024_Deconvolution (copy archived at *Uzel, 2025*).

The following datasets were generated:

| Author(s) | Year | Dataset title | Dataset URL | Database and Identifier |
|---|---|---|---|---|
| Buzatu I, Arzate Mejia RG | 2023 | RNA-seq of mouse spermatogonia cells at three developmental stages (PND8, PND15 and adult) | https://www.ebi.ac.uk/biostudies/arrayexpress/studies/E-MTAB-12721 | ArrayExpress, E-MTAB-12721 |
| Buzatu I, Arzate Mejia RG | 2023 | ATAC-seq of mouse spermatogonia cells at two developmental stages (PND15 and adult) | https://www.ebi.ac.uk/biostudies/arrayexpress/studies/E-MTAB-12722 | ArrayExpress, E-MTAB-12722 |

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

# Appendix 1

**Appendix 1—key resources table**

| Reagent type (species) or resource | Designation | Source or reference | Identifiers | Additional information |
|---|---|---|---|---|
| Strain, strain background (*M. musculus*) | C57BL/6JRj (*M. musculus*) | Janvier Laboratories | Non applicable | - |
| Biological sample (*M. musculus*) | Primary spermatogonia (SPGs) | C57BL/6JRj males | Non applicable | Freshly isolated from pups and adult males |
| Antibody | BV421-conjugated anti-mouse B2 microglobulin (B2M; S19.8; Mouse monoclonal) | BD Biosciences | Cat# 744802 | Cell enrichment from pups and adult testes |
| Antibody | Biotin Rat Anti-Mouse CD90.2 (Thy-1.2; 53–2.1; Rat monoclonal) | BD Biosciences | Cat# 553002 | Cell enrichment from pups testes |
| Antibody | PE Rat Anti-mouse CD51 (integrin a5; Clone RMV-7; Rat monoclonal) | BD Biosciences | Cat#: 551187 | Cell enrichment from pups testes |
| Antibody | Streptavidin Alexa Fluor 488 conjugate | Invitrogen | Cat#: S32354 | Cell enrichment from pups and adult testes |
| Antibody | Rat Anti-Human CD49f (integrin a6; GoH3; Rat monoclonal) | BD Biosciences | Cat#: 555734 | Cell enrichment from adult testes |
| Antibody | PE Rat Anti-Mouse CD90.2 (Thy-1.2; Clone 30-H12; Rat monoclonal) | BD Biosciences | Cat#: 553014 | Cell enrichment from adult testes |
| Antibody | Biotin Mouse Anti-Rat IgG1/2 a (Clone G28-5; Mouse monoclonal) | BD Biosciences | Cat#: 553880 | Cell enrichment from adult testes |
| Antibody | PLZF (2A9; Mouse monoclonal) | Active Motif | Cat#: 39988 | Immunocytochemistry |
| Chemical compound, drug | Propidium iodide | Sigma-Aldrich | Cat#:537060 | FACS |
| Peptide, recombinant protein | Collagenase type IV | Sigma-Aldrich | Cat#: C4-28-100MG | Cell enrichment from pups and adult testes |
| Commercial assay or kit | AllPrep DNA/RNA Micro Kit | Qiagen | Cat#: 80284 | RNA extraction |
| Commercial assay or kit | SMARTer Stranded Total RNA-seq Kit v3 | Takara Bio USA, Inc | Cat#: 634485 | RNA-seq library preparation |
| Commercial assay or kit | SMARTer RNA Unique Dual Index Kit | Takara Bio USA, Inc | Cat#: 634451 | RNA-seq library preparation |
| Commercial assay or kit | Tn5 transposase | Illumina | Cat#:15027916 | ATAC-seq library |
| Commercial assay or kit | Nextera XT Index Kit 24-indexes | Illumina | Cat#:15055293 | ATAC-seq library |
| Commercial assay or kit | MinElute Reaction Cleanup Kit | Qiagen | Cat#:28204 | ATAC-seq library |
| Commercial assay or kit | NEBNext High-Fidelity 2 X PCR Master Mix | New England Biolabs | Cat#:M0541S | ATAC-seq library |
| Other | AMPure XP reagent | Beckman Coulter Life Sciences | Cat#: A63880 | Library purification and clean-up |

*Appendix 1 Continued on next page*

*Appendix 1 Continued*

| Reagent type (species) or resource | Designation | Source or reference | Identifiers | Additional information |
|---|---|---|---|---|
| Software | Salmon | Salmon/Bioconda | Version 0.9.1 | RNA-seq analysis |
| Software | edgeR | edgeR/Bioconductor | Version 3.28.1 | RNA-seq analysis/ATAC-seq analysis |
| Software | limma | Limma/Bioconductor | Version 3.42.2 | RNA-seq analysis |
| Software | G:Profiler | https://biit.cs.ut.ee/gprofiler/gost | Do not apply | RNA-seq and ATAC-seq analysis |
| Software | CIBERSORTx | https://cibersortx.stanford.edu/ | Do not apply | RNA-seq deconvolution analysis |
| Software | Bowtie2 | Bowtie2/Bioconda | Version 2.3.5 | RNA-seq and ATAC-seq analysis |
| Software | deepTools | https://deeptools.readthedocs.io/en/develop/content/installation.html | Version 3.4.3 | RNA-seq and ATAC-seq analysis |
| Software | MACS2 | MACS2/Bioconda | Version 2.2.7.1 | ATAC-seq analysis |
| Software | ChIPpeakAnno | ChIPpeakAnno/Bioconductor | Version 4.4 | ATAC-seq analysis |
| Software | MEME | MEME/Bioconductor | Version 5.4.1 | ATAC-seq analysis |
| Software | Rsubread | Rsubread/Bioconductor | Version 2.0.1 | ATAC-seq analysis |
| Software | csaw | csaw/Bioconductor | Version 1.20.0 | ATAC-seq analysis |
| Software | HOMER | http://homer.ucsd.edu/homer/introduction/install.html | Version 4.11.1 | ATAC-seq analysis |

