## [Editor Report · eLife Assessment]

This **useful** study reports datasets on gene expression and chromatin accessibility profiles of spermatogonia at different postnatal ages in mice. Overall, the technical aspects of the sequencing analyses and computational/bioinformatics are **solid**. This study may be of interest to biomedical researchers working on male germline stem cells and male fertility.

---

## [Referee Report · Reviewer #1 (Public Review)]

Summary

This study was designed to investigate changes in gene expression and associated chromatin accessibility patterns in spermatogonia in mice at different postnatal stages from pups to adults. The objective was to describe dynamic changes in these patterns that potentially correlate with functional changes in spermatogonia as a function of development and reproductive maturation. The potential utility of this information is to serve as a reference against which similar data from animals subjected to various disruptive environmental influences can be compared.

Major Strengths and Weaknesses of the Methods and Results

A strength of the study is that it reviews previously published datasets describing gene expression and chromatin accessibility patterns in mouse spermatogonia. A weakness of the study is that it is not clear what new information is provided by the data provided that was not already known from previously published studies (see below). Specific weaknesses include the following...

- Terminology - In the Abstract and first part of the Introduction the authors use the generic term "spermatogonial cells" in a manner that seems to be referring primarily to spermatogonial stem cells (SSCs) but initially ignores the well-known heterogeneity among spermatogonia - particularly the fact that only a small proportion of developing spermatogonia become SSCs - and ONLY those SSCs and NOT other developing spermatogonia - support steady-state spermatogenesis by retaining the capacity to either self-renew or contribute to the differentiating spermatogenic lineage throughout the male reproductive lifespan. The authors eventually mention other types of developing male germ cells, but their description of prospermatogonial stages that precede spermatogonial stages is deficient in that M-prospermatogonia - which occur after PGCs but before T1-prospermatogonia - are not mentioned. This description also seems to imply that all T2-prospermatogonia give rise to SSCs which is far from the case. It is the case that prospermatogonia give rise to spermatogonia, but only a very small proportion of undifferentiated spermatogonia form the foundational SSCs and ONLY SSCs possess the capacity to either self-renew or give rise to sequential waves of spermatogenesis.

- Introduction - Statements regarding distinguishing transcriptional signatures in spermatogonia at different postnatal stages appear to refer to ALL subtypes of spermatogonia present at each stage collectively, thereby ignoring the well-known fact that there are distinct spermatogonial subtypes present at each postnatal stage and that some of those occur at certain stages but not at others. This brings into question the usefulness of the authors' discussion of what types of genes are expressed and/or what types of changes in chromatin accessibility are detected in spermatogonia at each stage.

- Methodology - The authors based recovery (enrichment) of spermatogonia from male pups on FACS sorting for THY1 and RMV-1. While sorting total testis cells for THY1+ cells does enrich for spermaogonia, this approach is now known to not be highly specific for spermatogonia (somatic cells are also recovered) and definitely not for SSCs. There are more effective means for isolating SSCs from total testis cells that have been validated by transplantation experiments (e.g. use of the Id4/eGFP transgene marker).

The authors then used "deconvolution" of bulk RNA-seq data in an attempt to discern spermatogonial subtype-specific transcriptomes. It is not clear why this is necessary or how it is beneficial given the availability of multiple single-cell RNA-seq datasets already published that accomplish this objective quite nicely - as the authors essentially acknowledge. Beyond this concern, a potential flaw with the deconvolution of bulk RNA-seq data is that this is a derivative approach that requires assumptions/computational manipulations of apparent mRNA abundance estimates that may confound interpretation of the relative abundance of different cellular subtypes within the hetergeneous cell population from which the bulk RNA-seq data is derived. Bottom line, it is not clear that this approach affords any experimental advantage over use of the publicly available scRNA-seq datasets and it is possible that attempts to employ this approach may be flawed yielding misleading data.

- Results & Discussion - In general, much of the information reported in this study is not novel. The authors' discussion of the makeup of various spermatogonial subtypes in the testis at various ages does not really add anything to what has been known for many years on the basis of classic morphological studies. Further, as noted above, the gene expression data provided by the authors on the basis of their deconvolution of bulk RNA-seq data does not add any novel information to what has been shown in recent years by multiple elegant scRNA-seq studies - and, in fact, as also noted above - represents an approach fraught with potential for misleading results. The potential value of the authors' report of "other cell types" not corresponding to major somatic cell types identified in earlier published studies seems quite limited given that they provide no follow-up data that might indicate the nature of these alternative cell types. Beyond this, much of the gene expression and chromatin accessibility data reported by the authors - by their own admission given the references they cite - is largely confirmatory of previously published results. Similarly, results of the authors' analyses of putative factor binding sites within regions of differentially accessible chromatin also appear to confirm previously reported results. Ultimately, it is not at all novel to note that changes in gene expression patterns are accompanied by changes in patterns of chromatin accessibility in either related promoters or enhancers. The discussion of these observations provided by the authors takes on more of a review nature than that of any sort of truly novel results. As a result, it is difficult to discern how the data reported in this manuscript advance the field in any sort of novel or useful way beyond providing a review of previously published studies on these topics.

Likely impact - The likely impact of this work is relatively low because, other than the value it provides as a review of previously published datasets, the new datasets provided are not novel and so do not advance the field in any significant manner.

---

## [Referee Report · Reviewer #2 (Public Review)]

This revised manuscript attempts to explore the underlying chromatin accessibility landscape of spermatogonia from the developing and adult mouse testis. The key criticism of the first version of this manuscript was that bulk preparations of mixed populations of spermatogonia were used to generate the data that form the basis of the entire manuscript. To address this concern, the authors applied a deconvolution strategy (CIBERSORTx (Newman et al., 2019)) in an attempt to demonstrate that their multi-parameter FACS isolation (from Kubota 2004) of spermatogonia enriched for PLZF+ cells recovered spermatogonial stem cells (SSCs). PLZF (ZBTB16) protein is a transcription factor known to mark all or nearly all undifferentiated spermatogonia and some differentiating spermatogonia (KIT+ at the protein level) - see Niedenberger et al., 2015 (PMID: 25737569). The authors' deconvolution using single-cell transcriptomes produced at postnatal day 6 (P6) argue that 99% of the PLZF+ spermatogonia at P8 are SSCs, 85% at P15 and 93% in adults. Quite frankly given the established overlap between PLZF and KIT and known identity of spermatogonia at these developmental stages, this is impossible. Indeed - the authors' own analysis of the reference dataset demonstrates abundant PLZF mRNA in P6 progenitor spermatogonia - what is the authors' explanation for this observation? The same is essentially true in the use of adult references for celltype assignment. The authors found 63-82% of SSCs using this different definition of types (from a different dataset), begging the question of which of these results is true.

In their rebuttal, the authors also raise a fair point about the precision of differential gene expression among spermatogonial subsets. At the mRNA level, Kit is definitely detectable in undifferentiated spermatogonia, but it is never observed at the protein level until progenitors respond to retinoic acid (see Hermann et al., 2015). I agree with the authors that the mRNAs for "cell type markers" are rarely differentially abundant at absolute levels (0 or 1), but instead, there are a multitude of shades of grey in mRNA abundance that "separate" cell types, particularly in the male germline and among the highly related spermatogonial subtypes of interest (SSCs, progenitor spermatogonia and differentiating spermatogonia). That is, spermatogonial biology should be considered as a continuous variable (not categorical), so examining specific cell populations with defined phenotypes (markers, function) likely oversimplifies the underlying heterogeneity in the male germ lineage. But, here, the authors have ignored this heterogeneity entirely by selecting complex populations and examining them in aggregate. We already know that PLZF protein marks a wide range of spermatogonia, complicating the interpretation of aggregate results emerging from such samples. In their rebuttal, the authors nicely demonstrate the existence of these mixtures using deconvolution estimation. What remains a mystery is why the authors did not choose to perform single-cell multiome (RNA-seq + ATAC-seq) to validate their results and provide high-confidence outcomes. This is an accessible technique and was requested after the initial version, but essentially ignored by the authors.

A separate question is whether these data are novel. A prior publication by the Griswold lab (Schleif et al., 2023; PMID: 36983846) already performed ATAC-seq (and prior data exist for RNA-seq) from germ cells isolated from synchronized testes. These existing data are higher resolution than those provided in the current manuscript because they examine germ cells before and after RA-induced differentiation, which the authors do not base on their selection methods. Another prior publication from the Namekawa lab extensively examined the transcriptome and epigenome in adult testes (Maezawa et al., 2000; PMID: 32895557; and several prior papers). The authors should explain how their results extend our knowledge of spermatogonial biology in light of the preceding reports.

The authors are also encouraged to improve their use of terminology to describe the samples of interest. The mitotic male germ cells in the testis are called spermatogonia (not spermatogonial cells, because spermatogonia are cells). Spermatogonia arise from Prospermatogonia. Spermatogonia are divisible into two broad groups: undifferentiated spermatogonia (comprised of few spermatogonial stem cells or SSCs and many more progenitor spermatogonia - at roughly 1:10 ratio) and differentiating spermatogonia that have responded to RA. The authors also improperly indicate that SSCs directly produce differentiating spermatogonia - indeed, SSCs produce transit-amplifying progenitor spermatogonia, which subsequently differentiate in response to retinoic acid stimulation. Further, the use of Spermatogonial cells (and SPGs) is imprecise because these terms do not indicate which spermatogonia are in question. Moreover, there have been studies in the literature which have used similar terms inappropriately to refer to SSCs, including in culture. A correct description of the lineage and disambiguation by careful definition and rigorous cell type identification would benefit the reader.

Overall, my concern from the initial version of this manuscript stands - critical methodological flaws prevent interpretation of the results and the data are not novel. Readers should take note that results in essentially all Figures do not reflect the biology of any one type of spermatogonium.

---

## [Referee Report · Reviewer #3 (Public Review)]

In this study, Lazar-Contes and colleagues aimed to determine whether chromatin accessibility changes in the spermatogonial population during different phases postnatal mammalian testis development. Because actions of the spermatogonial population set the foundation for continual and robust spermatogenesis and the gene networks regulating their biology are undefined, the goal of the study has merit. To advance knowledge, the authors used mice as a model and isolated spermatogonia from three different postnatal developmental age points using cell sorting methodology that was based on cell surface markers reported in previous studies and then performed bulk RNA-sequencing and ATAC-sequencing. Overall, the technical aspects of the sequencing analyses and computational/bioinformatics seems sound but there are several concerns with the cell population isolated from testes and lack of acknowledgement for previous studies that have also performed ATAC-sequencing on spermatogonia of mouse and human testes. The limitations, described below, call into question validity of the interpretations and reduce the potential merit of the findings.

I suggest changing the acronym for spermatogonial cells from SC to SPG for two reasons. First, SPG is the commonly used acronym in the field of mammalian spermatogenesis. Second, SC is commonly used for Sertoli Cells.

The authors should provide a rationale for why they used postnatal day 8 and 15 mice.

The FACS sorting approach used was based on cell surface proteins that are not germline specific so there was undoubtedly somatic cells in the samples used for both RNA and ATAC sequencing. Thus, it is essential to demonstrate the level of both germ cell and undifferentiated spermatogonial enrichment in the isolated and profiled cell populations. To achieve this, the authors used PLZF as a biomarker of undifferentiated spermatogonia. Although PLZF is indeed expressed by undifferentiated spermatogonia, there have been several studies demonstrating that expression extends into differentiating spermatogonia. In addition, PLZF is not germ cell specific and single cell RNA-seq analyses of testicular tissue has revealed that there are somatic cell populations that express Plzf, at least at the mRNA level. For these reasons, I suggest that the authors assess the isolated cell populations using a germ cell specific biomarker such as DDX4 in combination with PLZF to get a more accurate assessment of the undifferentiated spermatogonial composition. This assessment is essential for interpretation of the RNA-seq and ATAC-seq data that was generated.

A previous study by the Namekawa lab (PMID: 29126117) performed ATAC-seq on a similar cell population (THY1+ FACS sorted) that was isolated from pre-pubertal mouse testes. It was surprising to not see this study referenced to in the current manuscript. In addition, it seems prudent to cross-reference the two ATAC-seq datasets for commonalities and differences. In addition, there are several published studies on scATAC-seq of human spermatogonia that might be of interest to cross-reference with the ATAC-seq data presented in the current study to provide an understanding of translational merit for the findings.

---

## [Author Response]

The following is the authors’ response to the current reviews.

**Reviewer #1 (Public Review):**
Summary - This study was designed to investigate changes in gene expression and associated chromatin accessibility patterns in spermatogonia in mice at different postnatal stages from pups to adults. The objective was to describe dynamic changes in these patterns that potentially correlate with functional changes in spermatogonia as a function of development and reproductive maturation. The potential utility of this information is to serve as a reference against which similar data from animals subjected to various disruptive environmental influences can be compared.Major Strengths and Weaknesses of the Methods and Results - A strength of the study is that it reviews previously published datasets describing gene expression and chromatin accessibility patterns in mouse spermatogonia. A weakness of the study is that it is not clear what new information is provided by the data provided that was not already known from previously published studies (see below). Specific weaknesses include the following:Terminology - in the Abstract and first part of the Introduction the authors use the generic term "spermatogonial cells" in a manner that seems to be referring primarily to spermatogonial stem cells (SSCs) but initially ignores the well-known heterogeneity among spermatogonia - particularly the fact that only a small proportion of developing spermatogonia become SSCs - and ONLY those SSCs and NOT other developing spermatogonia - support steady-state spermatogenesis by retaining the capacity to either self-renew or contribute to the differentiating spermatogenic lineage throughout the male reproductive lifespan. The authors eventually mention other types of developing male germ cells, but their description of prospermatogonial stages that precede spermatogonial stages is deficient in that M-prospermatogonia - which occur after PGCs but before T1-prospermatogonia - are not mentioned. This description also seems to imply that all T2-prospermatogonia give rise to SSCs which is far from the case. It is the case that prospermatogonia give rise to spermatogonia, but only a very small proportion of undifferentiated spermatogonia form the foundational SSCs and ONLY SSCs possess the capacity to either self-renew or give rise to sequential waves of spermatogenesis.

We thank Reviewer 1 for the comments and clarifications. As suggested in the previous revision, we use the term spermatogonial cells (SPGs) to make it clear that our cell preparations do not exclusively contain SSCs but all SPGs since they derive from a FACS enrichment strategy. This is explained in the manuscript. Further, we conducted deconvolution analyses on the datasets to examine the composition of the enriched SPGs preparations and provide new sequencing information confirming the presence of SSCs and differentiating SPGs.

Introduction - Statements regarding distinguishing transcriptional signatures in spermatogonia at different postnatal stages appear to refer to ALL subtypes of spermatogonia present at each stage collectively, thereby ignoring the well-known fact that there are distinct spermatogonial subtypes present at each postnatal stage and that some of those occur at certain stages but not at others. This brings into question the usefulness of the authors' discussion of what types of genes are expressed and/or what types of changes in chromatin accessibility are detected in spermatogonia at each stage.

We agree that our data do not provide information about the transcriptional program of each subtype of SPGs. Rather they provide information about the dynamics of transcriptional programs in the transition from postnatal stage to adulthood in an enriched population of SPGs. The datasets are comprehensive and contain mRNA and non-coding RNA (with and without a polyA+ tail), which provides more precise transcriptomic information than classical single cell methods.

Methodology - The authors based recovery (enrichment) of spermatogonia from male pups on FACS sorting for THY1 and RMV-1. While sorting total testis cells for THY1+ cells does enrich for spermaogonia, this approach is now known to not be highly specific for spermatogonia (somatic cells are also recovered) and definitely not for SSCs. There are more effective means for isolating SSCs from total testis cells that have been validated by transplantation experiments (e.g. use of the Id4/eGFP transgene marker).

We acknowledge the technical limitations of our enrichment strategy and made them clear in our revised manuscript.

The authors then used "deconvolution" of bulk RNA-seq data in an attempt to discern spermatogonial subtype-specific transcriptomes. It is not clear why this is necessary or how it is beneficial given the availability of multiple single-cell RNA-seq datasets already published that accomplish this objective quite nicely - as the authors essentially acknowledge. Beyond this concern, a potential flaw with the deconvolution of bulk RNA-seq data is that this is a derivative approach that requires assumptions/computational manipulations of apparent mRNA abundance estimates that may confound interpretation of the relative abundance of different cellular subtypes within the hetergeneous cell population from which the bulk RNA-seq data is derived. Bottom line, it is not clear that this approach affords any experimental advantage over use of the publicly available scRNA-seq datasets and it is possible that attempts to employ this approach may be flawed yielding misleading data.

The deconvolution analyses were necessary to address the question of the cell composition of our preparations raised by reviewers. These analyses were highly beneficial because they clarify the presence of different SPGs including SSCs in the samples. They are also advantageous because the datasets they are conducted upon have significantly higher sequencing coverage than published single cell datasets. They contain the full transcriptome and not just polyA+ transcripts as 10x datasets thus they provide considerably richer and more comprehensive transcriptomic information. This is very important to correctly interpret the results and to gain additional biological information. For the deconvolution analyses, we used state-of-the-art methods with proper computational controls for calibration. We selected published single-cell RNA-seq datasets of the highest quality. These analyses are extremely useful because they confirm the predominance of SSCs in the postnatal and adult cell samples and a minimal contamination by somatic cells. Our approach also provides a useful workflow that can easily be used by other researchers who cannot afford single-cell RNA-seq and allow them gain more information about the cellular composition of their samples. Finally, the execution of any computational analyses, including analyses of single-cell RNA-seq datasets requires to make assumptions during the development and the use of a method. The assumptions made for deconvolution analyses are not special in this respect and do not introduce more confounds than other methods. What is critical for such analyses is to include proper controls for calibration, which we carefully did and validated using our own previously published datasets for Sertoli cells.

Results & Discussion - In general, much of the information reported in this study is not novel. The authors' discussion of the makeup of various spermatogonial subtypes in the testis at various ages does not really add anything to what has been known for many years on the basis of classic morphological studies. Further, as noted above, the gene expression data provided by the authors on the basis of their deconvolution of bulk RNA-seq data does not add any novel information to what has been shown in recent years by multiple elegant scRNA-seq studies - and, in fact, as also noted above - represents an approach fraught with potential for misleading results. The potential value of the authors' report of "other cell types" not corresponding to major somatic cell types identified in earlier published studies seems quite limited given that they provide no follow-up data that might indicate the nature of these alternative cell types. Beyond this, much of the gene expression and chromatin accessibility data reported by the authors - by their own admission given the references they cite - is largely confirmatory of previously published results. Similarly, results of the authors' analyses of putative factor binding sites within regions of differentially accessible chromatin also appear to confirm previously reported results. Ultimately, it is not at all novel to note that changes in gene expression patterns are accompanied by changes in patterns of chromatin accessibility in either related promoters or enhancers. The discussion of these observations provided by the authors takes on more of a review nature than that of any sort of truly novel results. As a result, it is difficult to discern how the data reported in this manuscript advance the field in any sort of novel or useful way beyond providing a review of previously published studies on these topics.Likely impact - The likely impact of this work is relatively low because, other than the value it provides as a review of previously published datasets, the new datasets provided are not novel and so do not advance the field in any significant manner.

We acknowledge that much of the reported information is not novel but this is not necessarily a drawback as sequencing datasets on the same tissues or cells produced by different groups using comparable methods are common. This does not diminish the validity and usefulness of the datasets but rather enriches the respective fields as omics methods and data analyses can deliver different findings. Thus, our study cannot be criticized and disqualified because other datasets have been published but instead it should be acknowledged for providing high resolution full transcriptome information from different stages and adult of SCs that other studies do not provide. In this respect, the subjective nature of Reviewer 1’s statements is of concern. For instance, the statement: “…represents an approach fraught with potential for misleading results”. Such declaration suggests that all studies that previously used enrichment strategies are “fraught with potential for misleading results», which disqualifies the work of many colleagues. Further, this wrongly assumes that newer technologies are exempt of “potential for misleading results» which is not the case. Single-cell RNA-seq methods, extensively used to study SPGs, has been questioned for their limitation and potential biases due to low sequencing coverage, issues with transcript detection, low capture efficiency and higher degree of noise than bulk RNA datasets. Thus, caution is needed to interpret single-cell datasets on SPGs and these datasets also have their biases. For our datasets, we made major efforts to address the criticisms raised by the reviewer and reduce any potential misleading information by conducting additional analyses, by providing more details on the methods and enrichment strategy and by being careful with data interpretation. We would be grateful if these efforts could be acknowledged and the improvements on the manuscript and the value of the datasets be evaluated with objectivity.

**Reviewer #2 (Public Review):**
This revised manuscript attempts to explore the underlying chromatin accessibility landscape of spermatogonia from the developing and adult mouse testis. The key criticism of the first version of this manuscript was that bulk preparations of mixed populations of spermatogonia were used to generate the data that form the basis of the entire manuscript. To address this concern, the authors applied a deconvolution strategy (CIBERSORTx (Newman et al., 2019)) in an attempt to demonstrate that their multi-parameter FACS isolation (from Kubota 2004) of spermatogonia enriched for PLZF+ cells recovered spermatogonial stem cells (SSCs). PLZF (ZBTB16) protein is a transcription factor known to mark all or nearly all undifferentiated spermatogonia and some differentiating spermatogonia (KIT+ at the protein level) - see Niedenberger et al., 2015 (PMID: 25737569). The authors' deconvolution using single-cell transcriptomes produced at postnatal day 6 (P6) argue that 99% of the PLZF+ spermatogonia at P8 are SSCs, 85% at P15 and 93% in adults. Quite frankly given the established overlap between PLZF and KIT and known identity of spermatogonia at these developmental stages, this is impossible. Indeed - the authors' own analysis of the reference dataset demonstrates abundant PLZF mRNA in P6 progenitor spermatogonia - what is the authors' explanation for this observation? The same is essentially true in the use of adult references for celltype assignment. The authors found 63-82% of SSCs using this different definition of types (from a different dataset), begging the question of which of these results is true.

For full transparency, we provided information about the deconvolution analyses for all libraries that use cell-type specific matrices generated from PND6 and adult single-cell RNA-seq reference datasets in our previous response (Fig1-3, response to reviewer 1). However, we don’t claim “that 99% of the PLZF+ spermatogonia at P8 are SSCs, 85% at P15 and 93% in adults”. Of these percentages, the ones that correspond to our postnatal libraries are the ones reported in our updated manuscript (Please see FigS2). Importantly, we never claimed that these percentages correspond to “PLZF+ spermatogonia», exclusively. Rather, they were inferred using gene expression-specific signature matrices (Fig1-c response to Reviewer 1 as example). As clearly evident in feature maps in FigS2 of our updated manuscript, the cellular population identified as SSCs using the dataset from Hermann et al., 2018 shows overlap for the expression of Ddx4, Zbtb16 (PLZF), Gfra1 and Id4 but minimal Kit. In agreement with the reviewer’s observation, progenitors also show a signal for Zbtb16 but have a different gene expression signature matrix (see Fig.1c and 2c for an example of gene signature matrices from PND6 and adult samples from the same publication).

Regarding the question of which of these results are true, we observed that deconvolution analyses of our postnatal libraries using two different single-cell postnatal RNA-seq reference datasets consistently suggest a high contribution (>90%) by SSCs defined using cell-specific expression matrices following identification of cell-types that match the closest ones reported by each study (See FigS2 updated manuscript). The analyses of our adult libraries using published adult datasets from the same group (Hermann et al., 2018); Fig1 response to Reviewer 1 and FigS2 updated manuscript suggest that the contribution of adult SSCs to the cell population is lower than at postnatal stages, but SSCs still are the most abundant cell stage identified in our libraries (FigS2g). We reported these analyses and acknowledge that in our adult samples, we also likely have differentiating SPGs.

In their rebuttal, the authors also raise a fair point about the precision of differential gene expression among spermatogonial subsets. At the mRNA level, Kit is definitely detectable in undifferentiated spermatogonia, but it is never observed at the protein level until progenitors respond to retinoic acid (see Hermann et al., 2015). I agree with the authors that the mRNAs for "cell type markers" are rarely differentially abundant at absolute levels (0 or 1), but instead, there are a multitude of shades of grey in mRNA abundance that "separate" cell types, particularly in the male germline and among the highly related spermatogonial subtypes of interest (SSCs, progenitor spermatogonia and differentiating spermatogonia). That is, spermatogonial biology should be considered as a continuous variable (not categorical), so examining specific cell populations with defined phenotypes (markers, function) likely oversimplifies the underlying heterogeneity in the male germ lineage. But, here, the authors have ignored this heterogeneity entirely by selecting complex populations and examining them in aggregate. We already know that PLZF protein marks a wide range of spermatogonia, complicating the interpretation of aggregate results emerging from such samples. In their rebuttal, the authors nicely demonstrate the existence of these mixtures using deconvolution estimation. What remains a mystery is why the authors did not choose to perform single-cell multiome (RNA-seq + ATAC-seq) to validate their results and provide high-confidence outcomes. This is an accessible technique and was requested after the initial version, but essentially ignored by the authors.

We agree with the reviewer that the male germ lineage should be considered as a continuous variable and that examining specific cell populations with defined features oversimplifies its heterogeneity. Regarding the use of single-cell multiome (RNA-seq + ATAC-seq), we also agree that this technology can provide additional insight by integrating RNA and chromatin accessibility in the same cells. However, it is an refined method that is expensive, time consuming and requires human resources that are beyond our capacity for this project.

A separate question is whether these data are novel. A prior publication by the Griswold lab (Schleif et al., 2023; PMID: 36983846) already performed ATAC-seq (and prior data exist for RNA-seq) from germ cells isolated from synchronized testes. These existing data are higher resolution than those provided in the current manuscript because they examine germ cells before and after RA-induced differentiation, which the authors do not base on their selection methods. Another prior publication from the Namekawa lab extensively examined the transcriptome and epigenome in adult testes (Maezawa et al., 2000; PMID: 32895557; and several prior papers). The authors should explain how their results extend our knowledge of spermatogonial biology in light of the preceding reports.

Our data do extend previous studies because they provide high-resolution transcriptomic (full transcriptome) and chromatin accessibility profiling in postnatal and adult stages. They now also provide an approach for deconvolution analyses of bulk RNA datasets that can be of use to the community. Novelty in the field of omics is usually not a prime feature and it is common that datasets on the same tissues or cells be published by different groups using comparable methods and analyses.

The authors are also encouraged to improve their use of terminology to describe the samples of interest. The mitotic male germ cells in the testis are called spermatogonia (not spermatogonial cells, because spermatogonia are cells). Spermatogonia arise from Prospermatogonia. Spermatogonia are divisible into two broad groups: undifferentiated spermatogonia (comprised of few spermatogonial stem cells or SSCs and many more progenitor spermatogonia - at roughly 1:10 ratio) and differentiating spermatogonia that have responded to RA. The authors also improperly indicate that SSCs directly produce differentiating spermatogonia - indeed, SSCs produce transit-amplifying progenitor spermatogonia, which subsequently differentiate in response to retinoic acid stimulation. Further, the use of Spermatogonial cells (and SPGs) is imprecise because these terms do not indicate which spermatogonia are in question. Moreover, there have been studies in the literature which have used similar terms inappropriately to refer to SSCs, including in culture. A correct description of the lineage and disambiguation by careful definition and rigorous cell type identification would benefit the reader.Overall, my concern from the initial version of this manuscript stands - critical methodological flaws prevent interpretation of the results and the data are not novel. Readers should take note that results in essentially all Figures do not reflect the biology of any one type of spermatogonium.

We revised and improved the terminology wherever possible and also considering requests from other reviewers about terminology.

**Reviewer #3 (Public Review):**
In this study, Lazar-Contes and colleagues aimed to determine whether chromatin accessibility changes in the spermatogonial population during different phases postnatal mammalian testis development. Because actions of the spermatogonial population set the foundation for continual and robust spermatogenesis and the gene networks regulating their biology are undefined, the goal of the study has merit. To advance knowledge, the authors used mice as a model and isolated spermatogonia from three different postnatal developmental age points using cell sorting methodology that was based on cell surface markers reported in previous studies and then performed bulk RNA-sequencing and ATAC-sequencing. Overall, the technical aspects of the sequencing analyses and computational/bioinformatics seems sound but there are several concerns with the cell population isolated from testes and lack of acknowledgement for previous studies that have also performed ATAC-sequencing on spermatogonia of mouse and human testes. The limitations, described below, call into question validity of the interpretations and reduce the potential merit of the findings.I suggest changing the acronym for spermatogonial cells from SC to SPG for two reasons. First, SPG is the commonly used acronym in the field of mammalian spermatogenesis. Second, SC is commonly used for Sertoli Cells.

This was suggested in the previous review by Reviewer 1 and was modified in the revised version of the manuscript.

The authors should provide a rationale for why they used postnatal day 8 and 15 mice. The FACS sorting approach used was based on cell surface proteins that are not germline specific so there was undoubtedly somatic cells in the samples used for both RNA and ATAC sequencing. Thus, it is essential to demonstrate the level of both germ cell and undifferentiated spermatogonial enrichment in the isolated and profiled cell populations. To achieve this, the authors used PLZF as a biomarker of undifferentiated spermatogonia. Although PLZF is indeed expressed by undifferentiated spermatogonia, there have been several studies demonstrating that expression extends into differentiating spermatogonia. In addition, PLZF is not germ cell specific and single cell RNA-seq analyses of testicular tissue has revealed that there are somatic cell populations that express Plzf, at least at the mRNA level. For these reasons, I suggest that the authors assess the isolated cell populations using a germ cell specific biomarker such as DDX4 in combination with PLZF to get a more accurate assessment of the undifferentiated spermatogonial composition. This assessment is essential for interpretation of the RNA-seq and ATAC-seq data that was generated.A previous study by the Namekawa lab (PMID: 29126117) performed ATAC-seq on a similar cell population (THY1+ FACS sorted) that was isolated from pre-pubertal mouse testes. It was surprising to not see this study referenced to in the current manuscript. In addition, it seems prudent to cross-reference the two ATAC-seq datasets for commonalities and differences. In addition, there are several published studies on scATAC-seq of human spermatogonia that might be of interest to cross-reference with the ATAC-seq data presented in the current study to provide an understanding of translational merit for the findings.

These points have been addressed in our previous response and in the revised manuscript.

The following is the authors’ response to the original reviews.

**Reviewer #1:**
Weaknesses:There appears to be a lack of basic knowledge of the process of spermatogenesis. For instance, the statement that "During the first week of postnatal life, a population of SCs continues to proliferate to give rise to undifferentiated Asingle (As), Apaired (Apr) and Aaligned (Aal) cells. The remaining SCs differentiate to form chains of daughter cells that become primary and secondary spermatocytes around postnatal day (PND) 10 to 12." is inaccurate. The Aal cells are the spermatogonial chains, the two are not distinct from one another. In addition, the authors fail to mention spermatogonial stem cells which form the basis for steady-state spermatogenesis. The authors also do not acknowledge the well-known fact that, in the mouse, the first wave of spermatogenesis is distinct from subsequent waves. Finally, the authors do not mention the presence of both undifferentiated spermatogonia (aka - type A) and differentiating spermatogonia (aka - type B). The premise for the study they present appears to be the implication that little is known about the dynamics of chromatin during the development of spermatogonia. However, there are published studies on this topic that have already provided much of the information that is presented in the current manuscript.

Regarding the inaccuracy and incompleteness of some of the statements about spermatogonial cells and spermatogenesis. In the Introduction, we replaced the following statement: "During the first week of postnatal life, a population of SCs continues to proliferate to give rise to undifferentiated Asingle (As), Apaired (Apr) and Aaligned (Aal) cells. The remaining SCs differentiate to form chains of daughter cells that become primary and secondary spermatocytes around postnatal day (PND) 10 to 12." by: “Spermatogonial cells (SPGs) are the initiators and supporting cellular foundation of spermatogenesis in testis in many species, including mammals. In the mammalian testis, the founding germ cells are primordial germ cells (PGCs), which give rise sequentially to different populations of SPGs : primary transitional (T1)-prospermatogonia (ProSG), secondary transitional (T2)-ProSG, and then spermatogonial stem cells (SSCs) (McCarrey, 2013; Rabbani et al., 2022; Tan et al., 2020). The ProSG population is exhausted by postnatal day (PND) 5 (Drumond et al., 2011) and by PND6-8, distinct SPGs subtypes can be distinguished on the basis of specific marker proteins and regenerative capacity (Cheng et al., 2020; Ernst et al., 2019; Green et al., 2018; Hermann et al., 2018; Tan et al., 2020).

SSCs represent an undifferentiated population of SPGs that retain regenerative capacity and divide to either self-renew or generate progenitors that initiate spermatogenic differentiation, giving rise to differentiating SPGs (diff-SPGs). Diff-SPGs form chains of daughter cells that become primary and secondary spermatocytes around PND10 to 12. Spermatocytes then undergo meiosis and give rise to haploid spermatids that develop into spermatozoa. Spermatozoa are then released into the lumen of seminiferous tubules and continue to mature in the epididymis until becoming capable of fertilization by PND42-48 in mice (Kubota and Brinster, 2018; Rooij, 2017).”

Regarding the premise and implications of our findings. We clarified the premise of our finding in the revised manuscript. The following statement was included in the Discussion: "our findings complement existing datasets on spermatogonial cells by providing parallel transcriptomic and chromatin accessibility maps at high resolution from the same cell populations at early postnatal, late postnatal and adult stages collected from single individuals (for adults)".

It is not clear which spermatogonial subtype the authors intended to profile with their analyses. On the one hand, they used PLZF to FACS sort cells. This typically enriches for undifferentiated spermatogonia. On the other hand, they report detection in the sorted population of markers such as c-KIT which is a well-known marker of differentiating spermatogonia, and that is in the same population in which ID4, a well-known marker of spermatogonial stem cells, was detected. The authors cite multiple previously published studies of gene expression during spermatogenesis, including studies of gene expression in spermatogonia. It is not at all clear what the authors' data adds to the previously available data on this subject.The authors analyzed cells recovered at PND 8 and 15 and compared those to cells recovered from the adult testis. The PND 8 and 15 cells would be from the initial wave of spermatogenesis whereas those from the adult testis would represent steady-state spermatogenesis. However, as noted above, there appears to be a lack of awareness of the well-established differences between spermatogenesis occurring at each of these stages.

We applied computational deconvolution to our bulk RNA-seq datasets, employing publicly available single-cell RNA-seq datasets, to estimate and identify cellular composition. Trained on high-quality RNA-seq datasets from pure or single-cell populations, deconvolution algorithms create expression matrices reflecting the cellular diversity in reference datasets. These cell-type-specific expression matrices are subsequently used to determine the cellular composition of bulk RNA-seq samples with unknown cellular components (Cobos et al., 2023).

For our analysis, we chose CIBERSORTx (Newman et al., 2019), recognized as the most advanced deconvolution algorithm to date, employing it with three high-quality, publicly available single-cell RNA-seq datasets. First, we assessed the cellular composition of all our RNA-seq libraries, using datasets generated by (Hermann et al., 2018) which characterized the single-cell transcriptomes of testicular cells and various populations of spermatogonial progenitor cells (SPGs) in early postnatal (PND6) and adult stages. This enabled us to not only address potential somatic cell contamination but also to analyse the composition of isolated SPGs using a unified dataset source.

**Author response image 1. sa4fig1:** Deconvolution analysis of bulk RNA-seq samples using PND6 single-cell RNA seq from Hermann et al, 2018. a.Seurat clusters from PND6 single-cell RNA-seq. b. Feature maps of gene expression for markers of SPGs and somatic cells. c. Gene expression signature matrix from PND6 single-cell RNA-seq datasets. d. Barplot of estimated cellular proportions for all bulk RNA-seq libraries reported in this study. e. Dotplot of the average estimated proportion of SSCs in all bulk RNA-seq libraries reported in this study.

By re-analyzing the single-cell RNA-seq datasets, we identified distinct cell-type clusters, marked by specific cellular markers as reported in the original and subsequent studies (Author response image 1a,b and Author response image 2a,b). Then, CIBERSORTx generated gene-expression signature matrices and estimated the cell-type proportions within our 18 bulk RNA-seq libraries. Evaluation of our postnatal libraries (PND8 and 15) against a PND6 signature matrix revealed a predominant derivation from SPGs, with average estimated proportions of spermatogonial stem cells (SSCs) being 0.99 and 0.85 for PND8 and PND15 samples, respectively (Author response image 1c-e). Notably, the analysis of PND15 libraries also suggested the presence of additional SPGs types, including progenitors and differentiating SPGs (Author response image 1d), albeit at lower frequency.

Similarly, evaluation of our adult RNA-seq libraries, using an adult signature matrix, showed an average SSC proportion of 0.82, indicating a primary derivation from SSC cells. Consistent with the findings from PND15 libraries, our deconvolution analysis also suggests the presence of additional SPG types, including progenitors and differentiating SPGs (Author response image 1d). However, unlike our early and late postnatal stage libraries, the deconvolution analysis of adult libraries indicated the presence of other cell types (labeled "Other"), not corresponding to the major somatic cell types identified by Hermann et al. 2018. The estimated average proportion of these cells was less than 0.05 in two adult libraries and 0.10 in the others. This variance in cellular composition underlines the deconvolution method's effectiveness in dissecting complex cellular compositions in bulk RNA-seq samples.

**Author response image 2. sa4fig2:** Deconvolution analysis of bulk RNA-seq samples using Adult single-cell RNA seq (Hermann et al, 2018). **(a)** Seurat clusters from Adult single-cell RNA-seq.

To further validate our observations, we re-analyzed two additional testicular single-cell RNA-seq datasets derived from an early postnatal stage (PND7) (Tan et al., 2020) and adult (Green et al., 2018) (Author response image 3a,b and Author response image 4a,b). We identified distinct cell-type clusters, marked by specific cellular markers (Author response image 3a,b and Author response image 4a,b), and proceeded with the deconvolution analysis using CIBERSORTx. Evaluation of our postnatal libraries (PND8 and 15) against the PND7 signature matrix from Tan et al., 2020 confirmed a derivation from germ cells (Author response image 3d,e), in particular from SSCs (Author response image 3g), with average estimated proportions of SSCs being 0.93 and 0.86 for PND8 and PND15 samples, respectively, and the rest estimated to be in origin from differentiating SPGs (Author response image 3g,h). In the case of the adult samples, evaluation against the adult signature matrix from Green et al., 2018 confirmed a predominant derivation from SSCs, with average estimated proportions of SSCs being 0.79, consistent with the 0.82 estimated proportion from Hermann et al., 2018.

**Author response image 3. sa4fig3:** Deconvolution analysis of bulk RNA-seq samples with additional single-cell datasets. Seurat clusters from PND7 single-cell RNA-seq (Tang 2020). b. Barplot of estimated cellular proportions for all bulk RNA-seq libraries reported in this study. c. Dotplot of the average estimated proportion of germ cells in all bulk RNA-seq libraries reported in this study. d. Re-clustering of germ cell cluster shown in a. e. Barplot of estimated cellular proportions for all bulk RNA-seq libraries reported in this study. f. Dotplot of the average estimated proportion of SSCs in all bulk RNA-seq libraries reported in this study. g. Seurat clusters from adult single-cell RNA-seq (Green et al., 2018). h. Barplot of estimated cellular proportions for all bulk RNA-seq libraries reported in this study. i. Dotplot of the average estimated proportion of germ cells in all bulk RNA-seq libraries reported in this study.

To further validate our deconvolution strategy, we interrogated the cellular composition of bulk RNA-seq libraries derived from cellular populations enriched in Sertoli cells, generated by our group using a similar enrichment/sorting strategy (Thumfart et al., 2022). As expected, our results show that all our libraries are mainly composed of Sertoli cells suggesting that the deconvolution strategy employed is accurate in detecting cell-type composition (Author response image 4).

**Author response image 4. sa4fig4:** Deconvolution analysis of Sertoli bulk RNA-seq samples. Barplots of estimated cellular proportions for bulk RNAseq libraries reported in Thumfart et al., 2022. Expression matrices were derived from the analysis of single-cell RNA-seq datasets used to asses cellular composition of the SPGs bulk libraries.

**Author response image 5. sa4fig5:** *Id4* and *Kit* are transcribed in SSCs. Seurat clusters from PND6 single-cell RNA-seq (left) and feature maps of gene expression for *Id4* (center) and *Kit* (right). Zoom in into SSCs (red).

Finally, regarding the following observation by the reviewer: "On the other hand, they report detection in the sorted population of markers such as c-KIT which is a well-known marker of differentiating spermatogonia, and that is in the same population in which ID4, a well-known marker of spermatogonial stem cells, was detected." It was recently shown using single-cell RNA that “nearly all differentiating spermatogonia at P3 (delineated as c-KIT+) are ID4-eGFP” (Law et al., 2019). While this finding does not exclude the fact that we have a mixture of SPGs cells, this finding supports the possibility that SPG cells express both markers of undifferentiated and differentiated cells, particularly in the early stages of postnatal development. Indeed, we observe that some cells labeled as SSC show signals for both *Id4* and *Kit* in single-cell RNA-seq data from Hermann et al., 2018 (Author response image 5).

Therefore, the results from the deconvolution analysis and our immunofluorescence data showing 85-95% PLZF+ cells in our cellular preparations underscore that our bulk RNA-seq libraries are mainly composed of SPGs. The deconvolution analysis also suggests a predominantly cellular composition of SSCs and to a lesser degree of differentiating SPGs. Our adult RNA-seq libraries show a small proportion of somatic cells (<0.10).

In the revised manuscript, we compiled the deconvolution analyses and present them in a condensed version in Supplementary Fig 2.

In general, the authors present observational data of the sort that is generated by RNA-seq and ATAC-seq analyses, and they speculate on the potential significance of several of these observations. However, they provide no definitive data to support any of their speculations. This further illustrates the fact that this study contributes little if any new information beyond that already available from the numerous previously published RNA-seq and ATAC-seq studies of spermatogenesis. In short, the study described in this manuscript does not advance the field.

We acknowledge that RNA-seq and ATAC-seq datasets like ours are observational and that their interpretation can be speculative. Nevertheless, our datasets represent an additional useful resource for the community because they are comprehensive and high resolution, and can be exploited for instance, for studies in environmental epigenetics and epigenetic inheritance examining the immediate and long-term effects of postnatal exposure and their dynamics. The depth of our RNA sequencing allowed detect transcripts with a high dynamic range, which has been limited with classical RNA sequencing analyses of spermatogonial cells and with single-cell analyses (which have comparatively low coverage). Further, our experimental pipeline is affordable (more than single cell sequencing approaches) and in the case of adults, provides data per animal informing on the intrinsic variability in transcriptional and chromatin regulation across males. These points will be discussed in the revised manuscript.

In general, the authors present observational data of the sort that is generated by RNA-seq and ATAC-seq analyses, and they speculate on the potential significance of several of these observations. However, they provide no definitive data to support any of their speculations. This further illustrates the fact that this study contributes little if any new information beyond that already available from the numerous previously published RNA-seq and ATAC-seq studies of spermatogenesis. In short, the study described in this manuscript does not advance the field.

Relevant information for both points was included in the Discussion of the revised manuscript.

The phenomenon of epigenetic priming is discussed, but then it seems that there is some expression of surprise that the data demonstrate what this reviewer would argue are examples of that phenomenon. The authors discuss the "modest correspondence between transcription and chromatin accessibility in SCs." Chromatin accessibility is an example of an epigenetic parameter associated with the primed state. The primed state is not fully equivalent to the actively expressing state. It appears that certain histone modifications along with transcription factors are critical to the transition between the primed and actively expressing states (in either direction). The cell types that were investigated in this study are closely related spermatogenic, and predominantly spermatogonial cell types. It is very likely that the differentially expressed loci will be primed in both the early (PND 8 or 15) and adult stages, even though those genes are differentially expressed at those stages. Thus, it is not surprising that there is not a strict concordance between +/- chromatin accessibility and +/- active or elevated expression.

Relevant information was included in the Discussion of the revised manuscript.

**Reviewer #2:**
The objective of this study from Lazar-Contes et al. is to examine chromatin accessibility changes in "spermatogonial cells" (SCs) across testis development. Exactly what SCs are, however, remains a mystery. The authors mention in the abstract that SCs are undifferentiated male germ cells and have self-renewal and differentiation activity, which would be true for Spermatogonial STEM Cells (SSCs), a very small subset of total spermatogonia, but then the methods they use to retrieve such cells using antibodies that enrich for undifferentiated spermatogonia encompass both undifferentiated and differentiating spermatogonia. Data in Fig. 1B prove that most (85-95%) are PLZF+, but PLZF is known to be expressed both by undifferentiated and differentiating (KIT+) spermatogonia (Niedenberger et al., 2015; PMID: 25737569). Thus, the bulk RNA-seq and ATAC-seq data arising from these cells constitute the aggregate results comprising the phenotype of a highly heterogeneous mixture of spermatogonia (plus contaminating somatic cells), NOT SSCs. Indeed, Fig. 1C demonstrates this by showing the detection of Kit mRNA (a well-known marker of differentiating spermatogonia - which the authors claim on line 89 is a marker of SCs!), along with the detection of markers of various somatic cell populations (albeit at lower levels).

The reviewer is correct that our spermatogonial cell populations are mixed and include undifferentiated and differentiated cells, hence the name of spermatogonia (SCs), and probably also contains some somatic cells. We acknowledge that this is a limitation of our isolation approach. To circumvent this limitation, we will conduct in silico deconvolution analysis using publicly available single-cell RNA sequencing datasets to obtain information about markers corresponding to undifferentiated and differentiated spermatogonia cells, and somatic cells. These additional analyses will provide information about the cellular composition of the samples and clarify the representation of undifferentiated and differentiated spermatogonial cells and other cells.

This admixture problem influences the results - the authors show ATAC-seq accessibility traces for several genes in Fig. 2E (exhibiting differences between P15 and Adult), including Ihh, which is not expressed by spermatogenic cells, and Col6a1, which is expressed by peritubular myoid cells. Thus, the methods in this paper are fundamentally flawed, which precludes drawing any firm conclusions from the data about changes in chromatin accessibility among spermatogonia (SCs?) across postnatal testis development.

The reviewer raises concern about the lack of correspondence between chromatin accessibility and expression observed for some genes, arguing that this precludes drawing firm conclusions. However, a dissociation between chromatin accessibility and gene expression is normal and expected since chromatin accessibility is only a readout of protein deposition and occupancy e.g. by transcription factors, chromatin regulators, or nucleosomes, at specific genomic loci that does not give functional information of whether there is ongoing transcriptional activity or not. A gene that is repressed or poised for expression can still show a clear signal of chromatin accessibility at regulatory elements. The dissociation between chromatin accessibility and transcription has been reported in many different cells and conditions (PMID: 36069349, PMID: 33098772) including in spermatogonial cells (PMID: 28985528) and in gonads in different species (PMID: 36323261). Therefore, the dissociation between accessibility and transcription is not a reason to conclude that our data are flawed.

In addition, there already are numerous scRNA-seq datasets from mouse spermatogenic cells at the same developmental stages in question.

This is true but full transcriptomic profiling like ours on cell populations provides different transcriptional information that is deeper and more comprehensive. Our datasets identified >17,000 genes while scRNA-seq typically identifies a few thousand of genes. Our analyses also identified full-length transcripts, variants, isoforms, and low abundance transcripts. These datasets are therefore a valuable addition to existing scRNAseq.

Moreover, several groups have used bulk ATAC-seq to profile enriched populations of spermatogonia, including from synchronized spermatogenesis which reflects a high degree of purity (see Maezawa et al., 2018 PMID: 29126117 and Schlief et al., 2023 PMID: 36983846 and in cultured spermatogonia - Suen et al., 2022 PMID: 36509798) - so this topic has already begun to be examined. None of these papers was cited, so it appears the authors were unaware of this work.

We apologize for not mentioning these studies in our manuscript, we will do so in the revised version.

The authors' methodological choice is even more surprising given the wealth of single-cell evidence in the literature since 2018 demonstrating the exceptional heterogeneity among spermatogonia at these developmental stages (the authors DID cite some of these papers, so they are aware). Indeed, it is currently possible to perform concurrent scATAC-seq and scRNA-seq (10x Genomics Multiome), which would have made these data quite useful and robust. As it stands, given the lack of novelty and critical methodological flaws, readers should be cautioned that there is little new information to be learned about spermatogenesis from this study, and in fact, the data in Figures 2-5 may lead readers astray because they do not reflect the biology of any one type of male germ cell. Indeed, not only do these data not add to our understanding of spermatogonial development, but they are damaging to the field if their source and identity are properly understood. Here are some specific examples of the problems with these data:Fig. 2D - Gata4 and Lhcgr are not expressed by germ cells in the testis.Fig. 3A - WT1 is expressed by Sertoli cells, so the change in accessibility of regions containing a WT1 motif suggests differential contamination with Sertoli cells. Since Wt1 mRNA was differentially high in P15 (Fig. 3B) - this seems to be the most likely explanation for the results. How was this excluded?Fig. 3D - Since Dmrt1 is expressed by Sertoli cells, the "downregulation" likely represents a reduction in Sertoli cell contamination in the adult, like the point above. Did the authors consider this?

Regarding concerns about contamination by somatic cells (Transcription). In addition to the results of our deconvolution analysis (see response to Reviewer #1), we addressed the specific concern of the *paradoxical expression* of genes considered markers of somatic cells in the testis. For instance, we plotted the expression values of *Ihh, Lhcgr, Gata4, Col16a, Wt1,* and *Dmrt1* along with the expression values of *Ddx4* and *Zbtb16*. We observe that the expression level of *Ddx4* and *Zbtb16*, genes expressed predominantly in SPGs, is orders of magnitude higher than the one observed for the rest of the genes with the notable exception of *Dmrt1* which is also highly expressed (Fig.6). Indeed, our analysis of publicly available single-cell RNA-seq datasets shows that *Dmrt1* is robustly expressed in germ cells (Author response image 7), and as also noted by the reviewer, in Sertoli cells in postnatal stages. Notably, we observe a significant stepwise decrease in the expression of *Dmrt1* across the postnatal maturation of SPG cells. This is highly unlikely to be a result of major contamination by Sertoli cells of just our postnatal libraries. We based this statement on three observations. First, the deconvolution analysis of all our RNA-seq libraries using four different expression signature matrices from high-quality single-cell RNAseq from testis showed that our libraries are largely derived from SPGs. Second, the evaluation of our adult libraries with the PND6 signature matrix from Green et al., 2018 suggested that the proportion of Sertoli cells in our adult libraries, if any, would be higher than in our postnatal libraries (Author response image 3d, blue bars). This makes it unlikely that the observed decrease in expression of *Dmrt1* in adult samples is due to prominent somatic contamination of the postnatal libraries. Third, the step-wise decrease in *Dmrt1* expression seems to correlate with progression during postnatal development (Author response image 7) as feature maps of *Dmrt1* expression derived from public single-cell RNA-seq experiments show a reduction in expression in adult SPGs in comparison with early postnatal stages (Author response image 7 last two panels). Then, the observed effects are likely the result of developmental gene regulatory processes that operate during the developmental maturation of SPGs.

**Author response image 6. sa4fig6:** Expression of germ and somatic cell markers in our RNA-seq datasets. Boxplots of log2(CPM) (Top) and CPM (Bottom) values for selected genes from our RNAseq datasets. Each point in boxplots represent the expression value of a biological replicate.

**Author response image 7. sa4fig7:** Expression of germ and somatic cell markers in publicly available single-cell RNA-seq datasets. Seurat clusters from all analyzed single-cell RNA-seq datasets (first column from left) and feature maps of gene expression for *Zbtb16*, *Dmrt1* and *Wt1.*

Consistent with the reviewer’s observation, *Ihh* is not expressed in germ cells and indeed we do not detect signal at this locus nor *Lhcgr*. Furthermore, while we indeed observe a significant increase in the expression of *Wt1* in PND15 samples, its expression level is considerably lower than that of SPG markers. This is even more evident when plotting expression data in a linear scale rather than as a log2 transformation of the expression values. Whether such transcriptional profiles reflect developmentally regulated transcription, stochastic effects on gene expression, or potential somatic contamination is difficult to determine. However, based on our deconvolution data we believe it is unlikely that major contamination could account for our observations.

Notably, while *Wt1* is robustly expressed in nearly all Sertoli cells across postnatal development (Author response image 7), it is also detected in other cell types including SPGs -although in fewer cells and with lower expression levels-, consistent with our observations (Author response image 6 and 8). Therefore, the assignment of a gene as a marker of a particular cell type does not imply that such a gene is expressed *uniquely* in such cell, rather it is expressed in more cells and likely at higher levels.

**Author response image 8. sa4fig8:** Expression of *Wt1* in publicly available single-cell RNA-seq datasets. Feature maps of gene expression for *Wt1.* In dashed boxes, a zoom-in into germ cells cluster that show expression of *Wt1* at some of these cells.

Regarding concerns about contamination by somatic cells (chromatin accessibility). In Figure 2 of our manuscript, we show the chromatin accessibility landscape of different genes, including genes either not expressed in testicular cells (*Ihh*) and those believed to be expressed exclusively in somatic cells (*Lhcgr, Gata4, Col16a1, Wt1*). For some of these genes, we reported changes in chromatin accessibility at specific sites between PND15 and adults (e.g. *Wt1* and *Col16a1*). The observation of "traces of chromatin accessibility" at these loci and the reported changes in accessibility raised concerns of potential contamination which "fundamentally flaw" our results, as stated by the reviewer. While we acknowledge that all enrichment methods have a margin of potential contamination, we fundamentally disagree with the reviewer's observations.

The term chromatin accessibility can be misleading. In principle, the term accessibility might suggest the literal lack of protein deposition at a given place in the genome. Rather, chromatin accessibility as evaluated by ATAC- seq (as in this case) must be interpreted as a measure of protein occupancy genome-wide (PMID: 30675018). Depending on the type of fragments analyzed we can obtain information regarding the occupancy of transcription factors (TFs), nucleosomes, and other chromatin-associated proteins that are present at genomic locations at a given time within a population of cells. The detection of chromatin accessibility at a given locus does not necessarily indicate transcription of the gene in a given cell type. A gene can be repressed or poised for expression and still show a clear signal of chromatin accessibility at its regulatory elements or along the gene body. For instance, in agreement with the reviewer's observation, neither *Ihh* nor *Lhcgr* is expressed in our datasets (Author response image 6 and Author response image 9), however, they show a distinctive pattern of chromatin accessibility in our datasets and publicly available ATAC-seq data derived from undifferentiated (Id4bright) and differentiating SPGs (Id4-dim) (Cheng et al., 2020) (Author response image 9). A similar argument can be applied regarding other loci such as *Wt1* and *Col6a1* for which we also observe extremely low levels of transcription. Therefore, the lack of transcription does not exclude that these loci display clear patterns of chromatin accessibility (Author response image 9). Notably, while traces of chromatin accessibility can also be observed in ATAC-seq datasets from embryonic Sertoli cells (Garcia-Moreno et al., 2019) and other somatic stem cells (hematopoietic stem cells; HSCs) (Xiang et al., 2020) (Author response image 9), the pattern of chromatin accessibility markedly differs with that observed in SPG cells. Therefore, the observed changes in chromatin accessibility are unlikely to result from contaminating somatic cells.

To strengthen our observation, we identified regions of chromatin accessibility in SPGs, Sertoli, and HSCs using both our datasets and publicly available ATAC-seq datasets. Overlap analysis revealed at least four groups of ATAC-seq peaks: (1) peaks shared among all analyzed cell types, (2)peaks shared just among SPG cells, (3) peaks specific to Sertoli cells and (4) peaks specific to HSCs (Author response image 10). Peaks shared among all tested cell-types are predominantly located at promoters of genes involved in translation and DNA replication (GO analysis adj p-value<0.05). In contrast, cell-type specific peaks are localized at intergenic and intragenic regions, suggesting localization at enhancer elements (Author response image 10). Indeed, GO analysis of cell-type specific peaks revealed enrichment for genes involved in male meiosis for SPGs, vesicle-mediated transport for Sertoli cells and in immune system process for HSCs, consistent with cell-type specific functions. If contamination by somatic cells, such as Sertoli cells, would be prominent as stated by the reviewer, we would expect to observe prominent ATAC-seq signal from our datasets at peaks specific to Sertoli cells. Notably, we don't observe ATAC-seq signal at peaks specific for Sertoli cells using our ATAC-seq samples. However, we observe robust signals at shared peaks and peaks specific to SPG cells. This observation, strongly argues against the possibility of major contamination by somatic cells.

**Author response image 9. sa4fig9:** Chromatin accessibility profiles at specific loci differ between SPG cells and other cell types. Genome-browser tracks for *Ihh*, *Wt1*, *Col16a1* and *Zbtb16*. For each gene, an extended locus view is presented with RNA-seq data (this study) and normalized ATAC-seq tracks from our study and public sources (SPG Id4; GSE131657; Sertoli; GSM3346484; HSC; ENCFF204JEE). Public ATAC-seq datasets were generated enrichment methods similar to the one employed in our study.

**Author response image 10. sa4fig10:** Shared and cell-type specific ATAC-seq peaks among SPGs, Sertoli and HSC.

**Reviewer #3:**
In this study, Lazar-Contes and colleagues aimed to determine whether chromatin accessibility changes in the spermatogonial population during different phases of postnatal mammalian testis development. Because actions of the spermatogonial population set the foundation for continual and robust spermatogenesis and the gene networks regulating their biology are undefined, the goal of the study has merit. To advance knowledge, the authors used mice as a model and isolated spermatogonia from three different postnatal developmental age points using a cell sorting methodology that was based on cell surface markers reported in previous studies and then performed bulk RNA-sequencing and ATAC-sequencing. Overall, the technical aspects of the sequencing analyses and computational/bioinformatics seem sound but there are several concerns with the cell population isolated from testes and lack of acknowledgment for previous studies that have also performed ATACsequencing on spermatogonia of mouse and human testes. The limitations, described below, call into question the validity of the interpretations and reduce the potential merit of the findings. I suggest changing the acronym for spermatogonial cells from SC to SPG for two reasons. First, SPG is the commonly used acronym in the field of mammalian spermatogenesis. Second, SC is commonly used for Sertoli Cells.

We thank the reviewer for the suggestion and will rename SCs into SPG cells in the revised manuscript.

The authors should provide a rationale for why they used postnatal day 8 and 15 mice.

We will provide a rationale for the use of postnatal 8 and 15 stages in the revised manuscript. Briefly, these stages are interesting to study because early to mid postnatal life is a critical window of development for germ cells during which environmental exposure can have strong and persistent effects. The possibility that changes in germ cells can happen during this period and persist until adulthood is an important area of research linked to disciplines like epigenetic toxicology and epigenetic inheritance.

The FACS sorting approach used was based on cell surface proteins that are not germline-specific so there were undoubtedly somatic cells in the samples used for both RNA and ATAC sequencing. Thus, it is essential to demonstrate the level of both germ cell and undifferentiated spermatogonial enrichment in the isolated and profiled cell populations. To achieve this, the authors used PLZF as a biomarker of undifferentiated spermatogonia. Although PLZF is indeed expressed by undifferentiated spermatogonia, there have been several studies demonstrating that expression extends into differentiating spermatogonia. In addition, PLZF is not germ-cell specific and single-cell RNA-seq analyses of testicular tissue have revealed that there are somatic cell populations that express Plzf, at least at the mRNA level. For these reasons, I suggest that the authors assess the isolated cell populations using a germ-cell specific biomarker such as DDX4 in combination with PLZF to get a more accurate assessment of the undifferentiated spermatogonial composition. This assessment is essential for the interpretation of the RNA-seq and ATAC-seq data that was generated.

In agreement with the reviewer’s observation, *Zbtb16* (PLZF) is expressed in germ cells but also in somatic cells, in particular in the dataset derived from Green et al., 2018 (Author response image 11). However, when evaluating the expression patterns of *Ddx4,* we noticed that similar to *Zbtb16*, it is expressed both in the germ line and in the somatic compartment (Author response image 11). Notably, we observe expression of *Ddx4* in SSC but also in progenitors and differentiating SPGs (Author response image 11g). These observations suggest that at least at the transcript level, both genes are transcribed in germ cells and to a lesser degree in somatic cells.

**Author response image 11. sa4fig11:** Single-cell expression of *Ddx4* and *Zbtb16*. Seurat clusters from all analyzed single-cell RNA-seq datasets (a,c,e,g,i) and feature maps of gene expression for *Ddx4* and *Zbtb16* (b,d,f,j, h).

Finally, our deconvolution analysis using geneexpression signature matrices for different cellular populations suggest that our RNA-seq and ATAC-seq libraries are largely derived from SPG cells and in particular of SSCs.

Furthermore, while this analysis suggested the presence of somatic cells, their proportion is minimal in comparison with germ cells (Author response images 1-4). This is also supported by ATAC-seq analysis of somatic cells from testis (Author response images 9 and 10).

A previous study by the Namekawa lab (PMID: 29126117) performed ATAC-seq on a similar cell population (THY1+ FACS sorted) that was isolated from pre-pubertal mouse testes. It was surprising to not see this study referenced in the current manuscript. In addition, it seems prudent to cross-reference the two ATAC-seq datasets for commonalities and differences. In addition, there are several published studies on scATACseq of human spermatogonia that might be of interest to cross-reference with the ATAC-seq data presented in the current study to provide an understanding of translational merit for the findings.

We compared our ATAC-seq datasets with the ones from (Maezawa et al., 2017) and those from (Cheng et al., 2020). All these datasets were generated from FACSs sorted cells enriched for undifferentiating and differentiating SPGs. Sequencing files from Cheng et al, 2020 were equally processed as described in out methods section, while our pipeline was adjusted to process files from Maezawa et al., 2018 as they were single-end sequencing files. We generated a reference set of peaks from SPGs and calculated signal scores for all peaks across all samples. Then, calculated the Pearson correlation for all pairwise comparisons and generated a heatmap of correlations (Author response image 12). Two clusters emerge that separate the SPG samples from the pachytene spermatocytes and round spermatids reported by Maezawa et al., 2018. As expected SPG samples clustered together based on study of origin. Consistently, our postnatal samples formed one cluster next to but separated from the adult one. Similarly, the id4-bright samples clustered together and next to the id4-sim and the sample applied for the Thy1 and cKit samples. Notably, our samples and the ones from Cheng et al., 2020 have a higher correlation with each other when compared with the ones from Maezawa et al., 2018. Given the fundamental difference in library sequencing (single-end instead of the widely used paired-end for ATAC-seq experiments) we reasoned a comparison with the Maezawa et al., 2018 datasets is not optimal. Therefore, this data in addition to the one presented before (see response to Reviewer 1 and 2) strongly supports a predominantly SPG derivation of all our sequencing libraries.

**Author response image 12. sa4fig12:** Pearson correlation at the peak level among different ATAC-seq datasets. a) Our ATAC-seq libraries and ATAC-seq libraries from b) Cheng et al., 2020 and c) Maezawa et al., 2020. Thy1-1 and cKit libraries correspond to undifferentiated and differentiating SPGs, respectively. PS, pachytene spermatocytes and RS, round spermatids. Correlation analysis was done using Deeptools.

References

Cheng K, Chen I-C, Cheng C-HE, Mutoji K, Hale BJ, Hermann BP, Geyer CB, Oatley JM, McCarrey JR. 2020. Unique Epigenetic Programming Distinguishes Regenerative Spermatogonial Stem Cells in the Developing Mouse Testis. *iScience* 23:101596. doi:10.1016/j.isci.2020.101596

Cobos FA, Panah MJN, Epps J, Long X, Man T-K, Chiu H-S, Chomsky E, Kiner E, Krueger MJ, Bernardo D di, Voloch L, Molenaar J, Hooff SR van, Westermann F, Jansky S, Redell ML, Mestdagh P, Sumazin P. 2023. Effective methods for bulk RNA-seq deconvolution using scnRNA-seq transcriptomes. *Genome Biol* 24:177. doi:10.1186/s13059-023-03016-6

Drumond AL, Meistrich ML, Chiarini-Garcia H. 2011. Spermatogonial morphology and kinetics during testis development in mice: a high-resolution light microscopy approach. *Reproduction* 142:145–155. doi:10.1530/rep-10-0431

Ernst C, Eling N, Martinez-Jimenez CP, Marioni JC, Odom DT. 2019. Staged developmental mapping and X chromosome transcriptional dynamics during mouse spermatogenesis. *Nat Commun* 10:1251. doi:10.1038/s41467-019-09182-1

Garcia-Moreno SA, Futtner CR, Salamone IM, Gonen N, Lovell-Badge R, Maatouk DM. 2019. Gonadal supporting cells acquire sex-specific chromatin landscapes during mammalian sex determination. *Dev Biol* 446:168–179. doi:10.1016/j.ydbio.2018.12.023

Green CD, Ma Q, Manske GL, Shami AN, Zheng X, Marini S, Moritz L, Sultan C, Gurczynski SJ, Moore BB, Tallquist MD, Li JZ, Hammoud SS. 2018. A Comprehensive Roadmap of Murine Spermatogenesis Defined by Single-Cell RNA-Seq. *Dev Cell* 46:651-667.e10. doi:10.1016/j.devcel.2018.07.025

Hermann BP, Cheng K, Singh A, Cruz LR-DL, Mutoji KN, Chen I-C, Gildersleeve H, Lehle JD, Mayo M, Westernströer B, Law NC, Oatley MJ, Velte EK, Niedenberger BA, Fritze D, Silber S, Geyer CB, Oatley JM, McCarrey JR. 2018. The Mammalian Spermatogenesis Single-Cell Transcriptome, from Spermatogonial Stem Cells to Spermatids. *Cell Rep* 25:1650-1667.e8. doi:10.1016/j.celrep.2018.10.026

Kubota H, Brinster RL. 2018. Spermatogonial stem cells†. *Biol Reprod* 99:52–74. doi:10.1093/biolre/ioy077

Law NC, Oatley MJ, Oatley JM. 2019. Developmental kinetics and transcriptome dynamics of stem cell specification in the spermatogenic lineage. *Nat Commun* 10:2787. doi:10.1038/s41467-019-10596-0

Maezawa S, Yukawa M, Alavattam KG, Barski A, Namekawa SH. 2017. Dynamic reorganization of open chromatin underlies diverse transcriptomes during spermatogenesis. *Nucleic Acids Res* 46:gkx1052-. doi:10.1093/nar/gkx1052

McCarrey JR. 2013. Toward a More Precise and Informative Nomenclature Describing Fetal and Neonatal Male Germ Cells in Rodents1. *Biol Reprod* 89:Article 47, 1-9. doi:10.1095/biolreprod.113.110502

Newman AM, Steen CB, Liu CL, Gentles AJ, Chaudhuri AA, Scherer F, Khodadoust MS, Esfahani MS, Luca BA, Steiner D, Diehn M, Alizadeh AA. 2019. Determining cell type abundance and expression from bulk tissues with digital cytometry. *Nat Biotechnol* 37:773–782. doi:10.1038/s41587-019-0114-2

Rabbani M, Zheng X, Manske GL, Vargo A, Shami AN, Li JZ, Hammoud SS. 2022. Decoding the Spermatogenesis Program: New Insights from Transcriptomic Analyses. *Annu Rev Genet* 56:339–368.

doi:10.1146/annurev-genet-080320-040045

Rooij DG de. 2017. The nature and dynamics of spermatogonial stem cells. *Development* 144:3022–3030. doi:10.1242/dev.146571

Tan K, Song H-W, Wilkinson MF. 2020. Single-cell RNAseq analysis of testicular germ and somatic cell development during the perinatal period. *Development* 147:dev183251. doi:10.1242/dev.183251

Thumfart KM, Lazzeri S, Manuella F, Mansuy IM. 2022. Long-term effects of early postnatal stress on Sertoli cells. *Front Genet* 13:1024805. doi:10.3389/fgene.2022.1024805

Xiang G, Keller CA, Heuston EF, Giardine BM, An L, Wixom AQ, Miller A, Cockburn A, Sauria MEG, Weaver K, Lichtenberg J, Göttgens B, Li Q, Bodine D, Mahony S, Taylor J, Blobel GA, Weiss MJ, Cheng Y, Yue F, Hughes J, Higgs DR, Zhang Y, Hardison RC. 2020. An integrative view of the regulatory and transcriptional landscapes in mouse hematopoiesis. *Genome Res* 30:gr.255760.119. doi:10.1101/gr.255760.119